# Data-driven, participatory characterization of farmer varieties discloses teff breeding potential under current and future climates

**Aemiro Bezabih Woldeyohannes[1,2†], Sessen Daniel Iohannes[1†], Mara Miculan[1], Leonardo Caproni[1], Jemal Seid Ahmed[1], Kauê de Sousa[3,4], Ermias Abate Desta[2], Carlo Fadda[5], Mario Enrico Pè[1], Matteo Dell'Acqua[1]\***

[1]Center of Plant Sciences, Scuola Superiore Sant'Anna, Pisa, Italy; [2]Amhara Regional Agricultural Research Institute, Bahir Dar, Ethiopia; [3]Digital Inclusion, Bioversity International, Montpellier, France; [4]Department of Agricultural Sciences, Inland Norway University of Applied Sciences, Hamar, Norway; [5]Biodiversity for Food and Agriculture, Bioversity International, Nairobi, Kenya

**\*For correspondence:**
m.dellacqua@santannapisa.it

†These authors contributed equally to this work

**Competing interest:** The authors declare that no competing interests exist.

**Abstract** In smallholder farming systems, traditional farmer varieties of neglected and underutilized species (NUS) support the livelihoods of millions of growers and consumers. NUS combine cultural and agronomic value with local adaptation, and transdisciplinary methods are needed to fully evaluate their breeding potential. Here, we assembled and characterized the genetic diversity of a representative collection of 366 Ethiopian teff (*Eragrostis tef*) farmer varieties and breeding materials, describing their phylogenetic relations and local adaptation on the Ethiopian landscape. We phenotyped the collection for its agronomic performance, involving local teff farmers in a participatory variety evaluation. Our analyses revealed environmental patterns of teff genetic diversity and allowed us to identify 10 genetic clusters associated with climate variation and with uneven spatial distribution. A genome-wide association study was used to identify loci and candidate genes related to phenology, yield, local adaptation, and farmers' appreciation. The estimated teff genomic offset under climate change scenarios highlighted an area around lake Tana where teff cropping may be most vulnerable to climate change. Our results show that transdisciplinary approaches may efficiently propel untapped NUS farmer varieties into modern breeding to foster more resilient and sustainable cropping systems.

## Editor's evaluation

Teff (*Eragrostis tef*), a small-market domesticate native and commonly grown in Ethiopia and the Horn of Africa, is comprehensively characterized for genetic, ecological and phenotypic variation in this ambitious and interdisciplinary publication. Integration of small holder farmers in phenotyping the collection, with an emphasis on gender considerations, elevates the characterization of Ethiopian teff. This paper provides a solid foundation to accelerate teff breeding for a changing climate, and provides an excellent model for the characterization of novel and underused crops.

## Introduction

Large-scale, high-yielding cropping systems rely on a remarkable small set of crops. Approximately half of the global farming land is devoted to maize, wheat, rice, and soybean (*FAOSTAT, 2021*) and

**eLife digest** Small farms support the livelihoods of about two billion people worldwide. Smallholder farmers often rely on local varieties of crops and use less irrigation and fertilizer than large producers. But smallholdings can be vulnerable to weather events and climate change. Data-driven research approaches may help to identify the needs of farmers, taking into account traditional knowledge and cultural practices to enhance the sustainability of certain crops.

Teff is a cereal crop that plays a critical role in the culture and diets of Ethiopian communities. It is also a super food appreciated on international markets for its nutritional value. Rural smallholder farmers in Ethiopia rely on the crop for subsistence and income and make up the bulk of the country's agricultural system. Many grow local varieties with tremendous genetic diversity. Scientists, in collaboration with farmers, could tap that diversity to produce more productive or climate-resilient types of teff, both for national and international markets.

Woldeyohannes, Iohannes et al. produced the first large-scale genetic, agronomic and climatic study of traditional teff varieties. In the experiments, Woldeyohannes and Iohannes et al. sequenced the genomes of 366 Ethiopian teff varieties and evaluated their agronomic value in common gardens. The team collaborated with 35 local farmers to understand their preference of varieties and traits. They then conducted a genome-wide association study to assess the crops' productivity and their adaptations to local growing conditions and farmer preferences. Genetic changes that speed up teff maturation and flowering time could meet small farmers' needs to secure teff harvest. Woldeyohannes, Iohannes et al. also identified a region in Ethiopia, where local teff varieties may struggle to adapt to climate change. Genetic modifications may help the crop to adapt to frequent droughts that may be a common characteristic of future climates.

The experiments reveal the importance of incorporating traditional knowledge from smallholder farmers into data-driven crop improvement efforts considering genetics and climate science. This multidisciplinary approach may help to improve food security and protect local genetic diversity on small farms. It may also help to ensure that agricultural advances fairly and equitably benefit small farmers.

the overall composition of worldwide food systems is rather uniform (*Khoury et al., 2014*). Yet, hundreds of neglected or underutilized species (NUS) are still actively cultivated in highly diversified, small-scale cropping systems, where they support the livelihoods of millions of people (*Jamnadass et al., 2020*). NUS benefited of scant research and breeding improvement. Their diversity is not only a proxy of pedoclimatic diversification of cropping systems, but also reflects socioeconomic diversity and cultural heritage of local farmers (*Tadele, 2019*). Rich NUS agrobiodiversity is still conserved *in situ* in smallholder farming systems, where the selection and cultivation conducted by local growers resulted in untapped local varieties that could provide useful adaptive traits (*Iragaba et al., 2020*). A comprehensive, transdisciplinary characterization of NUS farmer varieties that takes into consideration diversity, adaptation, and farmer–consumer preferences may unlock the potential of NUS for the sustainable intensification of farming systems (*Dawson et al., 2018*).

Crop scientists and breeders can now leverage the big data revolution to bridge the gap between NUS and 21st century agriculture (*Dawson et al., 2019*). Genomic tools enable a rapid and cost-effective characterization of large germplasm collections to unlock agrobiodiversity for breeding (*Poland, 2015*), identify genetic factors responsible for traits of agronomic interest (*Mascher et al., 2019*), and accelerate genetic gains (*Juliana et al., 2019*). Genomic data can be combined with climatic data to gain insights into locus-specific adaptation (*Lasky et al., 2015*) and to estimate genomic vulnerability under climate change scenarios (*Aguirre-Liguori et al., 2021*). Data-driven methods can also be applied to characterize the socioeconomic contexts in which crops are grown (*van Etten et al., 2019a*; *van Etten et al., 2019b*), generating insights that are critical to understand cropping dynamics in smallholder farming systems (*Gomez y Paloma et al., 2020*; *Iragaba et al., 2020*; *Terlau et al., 2019*). The agrobiodiversity maintained by farmers is the result of interactions between genetic, environmental, and societal factors, and has a large potential for varietal innovation when farmer preferences are factored into the selection process (*Mokuwa et al., 2014*). Participatory varietal selection (PVS) approaches have been developed to directly involve farmers in the evaluation

of breeding materials (*Ceccarelli and Grando, 2007*). Previous studies showed that data-driven PVS can be combined with genomic data to identify genomic loci responsible for farmers' appreciation (*Kidane et al., 2017*) and model local crop performance in farmer fields (*de Sousa et al., 2021*).

The Ethiopian highlands are a paradigm of challenging agricultural ecosystems where NUS farmer varieties are widely cultivated. In Ethiopia, 85 million people live in rural areas; most of them are subsistence-based smallholder farmers who are responsible for about 90% of the cultivated land and agricultural output of the country (*Bachewe and Taffesse, 2018*). Teff (*Eragrostis tef*) is an annual, self-pollinating, and allotetraploid grass of the *Chloridoideae* subfamily (*Ketema, 1997*). It is widely grown in Ethiopia as a staple crop, where it is valued for its nutritional and health benefits, resilience in marginal and semi-arid environments and cultural importance (*D'Andrea, 2008*; *Ketema, 1997*). Teff was likely domesticated in the northern Ethiopian Highlands from the allotetraploid *Eragrostis pilosa*, but the timing of its initial cultivation and the identity of its diploid ancestors remain poorly understood (*D'Andrea, 2008*; *Ingram and Doyle, 2003*; *VanBuren et al., 2020*). Nowadays, hundreds of local landraces are grown in Ethiopia in a wide range of agroecologies, displaying broad environmental adaptation and phenotypic diversity (*Woldeyohannes et al., 2020*). Breeding efforts on teff have been underway for decades, and segregant and mutagenized populations are available (*Cannarozzi et al., 2018*). Yet, teff yields today remain much lower than potentially attainable and substantially lower than those of other cereals grown in the region, and the full breeding potential of teff farmer varieties is still undisclosed (*Girma et al., 2014*; *Woldeyohannes et al., 2020*). Research in teff is rapidly evolving; a draft genome sequence (*Cannarozzi et al., 2014*) and a high-quality genome sequence (*VanBuren et al., 2020*) recently brought this species into the international genomics research spotlight. A data-driven research approach may unlock the full potential of teff agrobiodiversity and propel breeding for new improved varieties that meet local needs.

Here, we report a transdisciplinary data-driven approach to characterize NUS genetic, agronomic, and climatic diversity, using teff as a case study. We selected 321 teff farmer varieties derived from landraces and 45 teff improved lines, and we genotyped them with genome-wide molecular markers. Concurrently, we characterized their agronomic performance at two locations in Ethiopia. Experienced teff farmers (15 women and 20 men) were asked to evaluate the teff genotypes, providing qualitative information that enabled us to prioritize better adapted genetic materials. Additionally, we derived current and projected climate data at the sampling locations of teff accessions and used them to estimate genomic offset under climate change scenarios. We combined all these data in a genome-wide association study (GWAS) framework to identify genomic loci and candidate genes with relevance for adaptation, performance, and farmers' preferences. We discuss the potential of data-driven participatory approaches to characterize NUS diversity and to disclose their potential for the sustainable intensification of farming systems.

## Results and discussion
### Teff farmer varieties harness broad genetic diversity

We assembled a representative collection of teff cultivated in Ethiopia, hereafter named Ethiopian Teff Diversity Panel (EtDP). The EtDP comprises 321 farmer varieties spanning the entire geographical and agroecological range of teff cultivation in Ethiopia, from the submoist lowlands of Tigray in the North to the moist lowlands of Oromia in the South, and from the subhumid lowlands of Benishangul and Gumuz in the West to the subhumid mid-highlands of Oromia in the East (*Figure 1—figure supplement 1*, *Supplementary file 1A*; *MoA, 2000*). The EtDP also includes 45 improved varieties released since the first breeding efforts on teff and up to the moment of the EtDP assembly. A selection of seven *Eragrostis* spp., putative wild relatives of teff, was included as an outgroup. The genomic diversity of the EtDP was assessed starting from 12,153 high-quality, genome-wide single-nucleotide polymorphisms (SNPs) that were derived from double digest restriction site-associated sequencing (ddRAD-seq) of individual accessions, and were pruned for linkage disequilibrium (LD). The EtDP genomes could be grouped in 1240 haplotype blocks (*Supplementary file 1B*). EtDP chromosomes showed consistently higher pericentromeric LD, with localized LD peaks in telomeric regions (*Figure 1—figure supplement 2*). The A and B subgenomes showed comparable yet different LD profiles, possibly due to their specificity in terms of dominance, transposable elements content, and overall limited homoeologous exchange (*VanBuren et al., 2020*).

Population structure analysesshowed that EtDP accessions displayed varying degrees of genetic admixture (*Figure 1A*) and could be best summarized by 10 discriminant analysis of principal component (DAPC) clusters (*Figure 1—figure supplement 3*). About 15% of the genetic variability in the EtDP collection could be explained by the first three principal components (PCs) of LD-pruned SNP data (*Figure 1B, C*). Teff accessions from Tigray, in Northern Ethiopia, were markedly separated from the rest, and mainly belonged to cluster 7 (*Figure 1D*). Teff breeding lines showed a narrower genetic base compared to farmer varieties and could not summarize the broad diversity available within the EtDP (*Figure 1—figure supplement 4*). They belonged predominantly to clusters 2, 8, and 10 (*Figure 2D, E*). Genetic clusters are a proxy of teff landraces diversity and may support breeding efforts by improving the identification of parent lines for genomic selection to counter the depletion of allelic diversity (*Heffner et al., 2009*), or establishing breeding groups to explore heterotic potential in teff (*Boeven et al., 2016*). The preservation and valorization of teff genetic clusters are instrumental to respond to future breeding needs in light of changing climates and dynamic consumer preferences (*Woldeyohannes et al., 2020*; *Araya et al., 2011*).

## The distribution of teff genetic variation is associated with geographic and environmental factors

Landraces evolve at the interface of natural and anthropogenic selection (*Casañas et al., 2017*). Hence, we hypothesized that teff genetic clusters might be associated with local environmental conditions. The EtDP showed limited genetic stratification, as evidenced by the high levels of admixture and low genetic variance explained by the principal components analysis (PCA) (*Figure 1—figure supplement 5*; *Figure 1B, C*). Yet, some genetic clusters could be associated to agroecological zones characterized by different temperature and precipitation conditions (*Figure 1—figure supplement 6*; *Figure 3A, B*; *Figure 3—figure supplement 1*). Cluster 4 and 7 are cultivated in warmer sampling locations, while cluster 6 comes from colder and wetter areas (*Figure 3—figure supplement 1*). Extant landraces diversity is not only contributed by climate, but also by seed circulation. Studies have shown that cultural and social dynamics, such as belonging to similar or different ethnolinguistic groups, are key factors in shaping seed exchange networks (*Labeyrie et al., 2016*; *Samberg et al., 2007*). In Ethiopia, regional districts are markers of cultural and historical diversity, and smallholder farmers have limited capacity for long-distance travel. Thus, we hypothesized that regional distinctions could impact seed exchanges and spatial patterns of teff genetic diversity.

The district of sampling of teff landraces in the EtDP was used to aggregate accessions and calculate the fixation index ($F_{st}$) as a measure of genetic distances. Although our experimental design does not allow to fully untangle the effect of geography, environment, and social factors, we found a pattern of genetic variation distribution that could be associated to an isolation by distance and adaptation process, where $F_{st}$ values are significantly associated with geographic distance (Mantel $r$ = 0.31; p = 9e−04) and environmental distance (Mantel $r$ = 0.352; p = 0.0137; *Figure 3C, D*). Accessions from East Tigray (*Misraquawi*) showed the highest separation from the collection, followed by East Oromia (*Misraq Harerge*) and West Amhara (*Agew Awi*; *Figure 3E*). Tigray is believed to be the center of teff domestication, as earlier reports identified there much of its diversity (*Costanza et al., 2007*) and archaeobotanical excavations reported its cultivation there since the Pre Aksumite period (before I Century CE; *D'Andrea, 2008*). However, limited archeological information is available from other parts of the country, which may conceal other sites of early domestication of teff. When integrating the putative teff wild relatives *Eragrostis curvula* and *E. pilosa* in our teff landraces' phylogeny, we found that they did not group with any specific DAPC cluster (*Figure 3—figure supplement 2*).

## Participatory evaluation of the teff diversity prioritizes genetic materials for breeding

The EtDP was phenotyped for agronomic and farmer appreciation traits during the main cropping season in two locations in Ethiopia (*Figure 1D*) representing areas with high potential for teff cultivation. Experienced teff farmers including 15 women and 20 men provided their evaluations on overall appreciation (OA) and panicle appreciation (PA) across the EtDP. Women and men farmers were selected because they were both teff growers, and their responses were analyzed separately to highlight distinct trait evaluation criteria. Previous studies conducted in Ethiopia showed that agricultural traits relevant to marketability, food and drink preparations, animal feed, and construction are all

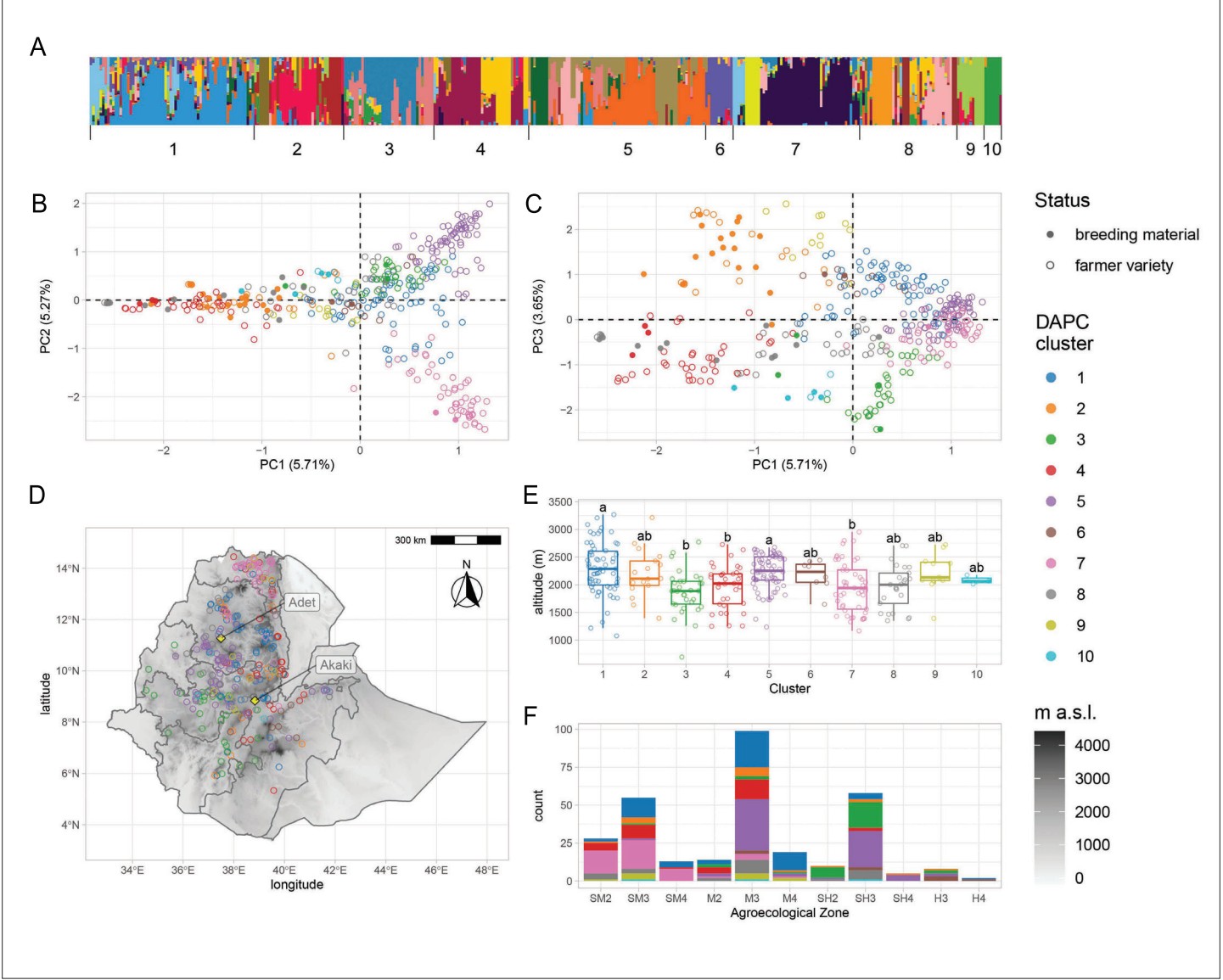

**Figure 1.** Genetic diversity of teff in Ethiopia. (**A**) ADMIXTURE results for the pruned single-nucleotide polymorphisms (SNPs) dataset. Each vertical bar represents an individual, colored according to one of the 20 groups reported by the analysis. Bars are ordered according to the 10 genetic clusters identified by discriminant analysis of principal component (DAPC), as reported by numbers on the x-axis. (**B, C**) Principal component analysis of genome-wide SNPs. Taxa are colored according to their DAPC genetic cluster, as indicated in the legend. About 10.98% of the genetic diversity in the panel can be explained by the first two principal components, which clearly separate cluster 7 from clusters 2 and 4. Open and close circles represent farmer varieties and improved varieties, respectively. (**D**) Distribution of Ethiopian Teff Diversity Panel (EtDP) georeferenced landraces (N = 314) across the altitudinal map of Ethiopia, color coded as in panel (**B**). (**E**) Altitude distribution across the DAPC genetic clusters,, with letters on top of boxplots denoting significance levels based on a pairwise Wilcoxon rank sum test and Bonferroni correction for multiple testing. (**F**) Distribution of genetic clusters across agroecological zones of Ethiopia, with color coding as in panel B. SM2, warm submoist lowlands; SM3, tepid submoist mid-highlands; SM4, cool submoist mid-highlands; M2, warm moist lowlands; M3, tepid moist mid-highlands; M4, cool moist mid-highlands; SH2, warm subhumid lowlands; SH3, tepid suphumid mid-highlands; SH4, cool subhumid mid-highlands; H3, tepid humid mid-highlands; H4, cool humid mid-highlands. This figure has six figure supplements.

The online version of this article includes the following figure supplement(s) for figure 1:

**Figure supplement 1.** Distribution of georeferenced farmer varieties in the Ethiopian Teff Diversity Panel (EtDP) (N = 314) overlaid to agroecological zones of Ethiopia.

**Figure supplement 2.** Linkage disequilibrium (LD) in subgenome A (**A**) and subgenome B (**B**).

**Figure supplement 3.** Predictive accuracy of the model-based unsupervised clustering (ADMIXTURE) and the discriminant analysis of principal components (DAPC) using the fivefold cross-validation procedure and the Bayesian information criterion (BIC), respectively.

*Figure 1 continued on next page*

**Figure supplement 5.** Population structure of the EtDP (*N* = 366).

**Figure supplement 6.** Residual plot for the Pearson's chi-squared test of independence between discriminant analysis of principal component (DAPC) genetic clusters and Ethiopian agroecological zones (df = 60, p value <2.2e−16).

perceived differently by women and men farmers (*Assefa et al., 2014*; *Mancini et al., 2017*). Regardless of gender, farmer evaluations were highly heritable and fully comparable to agronomic traits commonly targeted by breeding (*Supplementary file 1C*). OA provided by farmers across genders and across locations had a broad-sense heritability ($H^2$) of 0.81 (*Table 1*). $H^2$ of grain yield combined across the same two locations was 0.42 (*Table 2*). A significant portion of the variance for production traits could be explained by differences in location, including biomass (75%) and yield (67%). Conversely, phenology was mostly explained by genotype: this is the case of days to heading (92%) and days to maturity (86%; *Supplementary file 1C*). Depending on the trait, we found different effects of genetic background, location, and gender, a proxy of genotype by environment (G × E) interactions in determining teff performance and appreciation in tested locations. Genetic clusters had a significant effect on all traits, and location was important for yield and yield components (*Supplementary file 1D*). PA, plant height, panicle length, culm diameter, and panicle weight were affected by the interaction of location and genetic cluster. Gender and gender by location interactions always had significant effects for PVS traits (*Supplementary file 1D*).

The high $H^2$ achieved by farmers' OA may be because farmers, in providing their overall evaluation, not only consider yield but also yield components with high heritability (*Table 3*). Farmers' appreciation of teff genotypes was strongly associated with yield (p < 0.001) and its components regardless of gender, most notably plant height (p < 0.001) and grain filling period (GFP) (p < 0.05) (*Figure 2A*, *Figure 2—figure supplement 1*, *Table 3*). Plant height and GFP were very important for men, while women OA was highly correlated with biomass yield and panicle weight (*Table 3*). Different dynamics of trait prioritization by men and women are expected, and may reflect gender roles in the value chain (*Weltzien et al., 2019*). Studies across smallholder farming systems in Africa showed that gender-differentiated roles in agriculture may result in different trait preferences. In Ethiopian wheat, men are mostly endowed with field work, while women are concerned with marketability of grains (*Mancini et al., 2017*). Both in cassava (*Teeken et al., 2018*) and maize (*Voss et al., 2021*), women are mostly concerned with use traits, while men prioritize agronomic trats. The underlying causes of gender differentiation in teff trait preference need to be further characterized, also in relation to the multipurpose teff cultivation as food and feed. Still, men and women participating in the PVS were all expert teff growers, as reflected by the consistency of their evaluations (*Figure 2*). Previous studies showed that farmers' appreciation has a genetic basis in the evaluated crop, and may be used to perform genomic prediction (*de Sousa et al., 2021*) and identify genome-wide associations (*Kidane et al., 2017*).

The top ranking teff accessions according to men and women farmers captured different genetic backgrounds (*Figure 2B*), but the same trait combinations (*Figure 2C*). This suggested that farmers were consistently preferring the same teff ideotype regardless of their gender and regardless of the genetic background and geographic provenance of the accession. High yielding, high biomass, and fast maturing varieties were preferred. Individually, teff improved varieties showed high OA and high panicle length, which is a key component of OA (*Figure 2D, E*). This is supported by the fact that the evaluation fields were high-potential areas for teff cultivation. Genetic cluster 2, that is representative of most breeding materials, is associated with longer days to heading and maturity, higher plant heights and panicle lengths, greater number of total tillers and higher yields (*Figure 2—figure supplement 2*). Still, several traditional farmer varieties from different genetic backgrounds recorded comparable, at times superior, performance than improved lines (*Figure 2G*). By selecting farmer varieties that outperform improved varieties' performances in target breeding traits, it may be possible to prioritize landrace accessions for teff improvement (*Figure 2D, E*) or even immediately make them available to farmers, as suggested by previous experiences in wheat (*Fadda et al., 2020*). In areas exposed to terminal drought, common in Ethiopia, short maturation time is paramount to achieve harvestable yield (*Mengistu and Mekonnen, 2012*), and it is expectedly a major component of farmers' OA (*Figure 2A*). The time of maturation is therefore an obvious target trait for teff breeding,

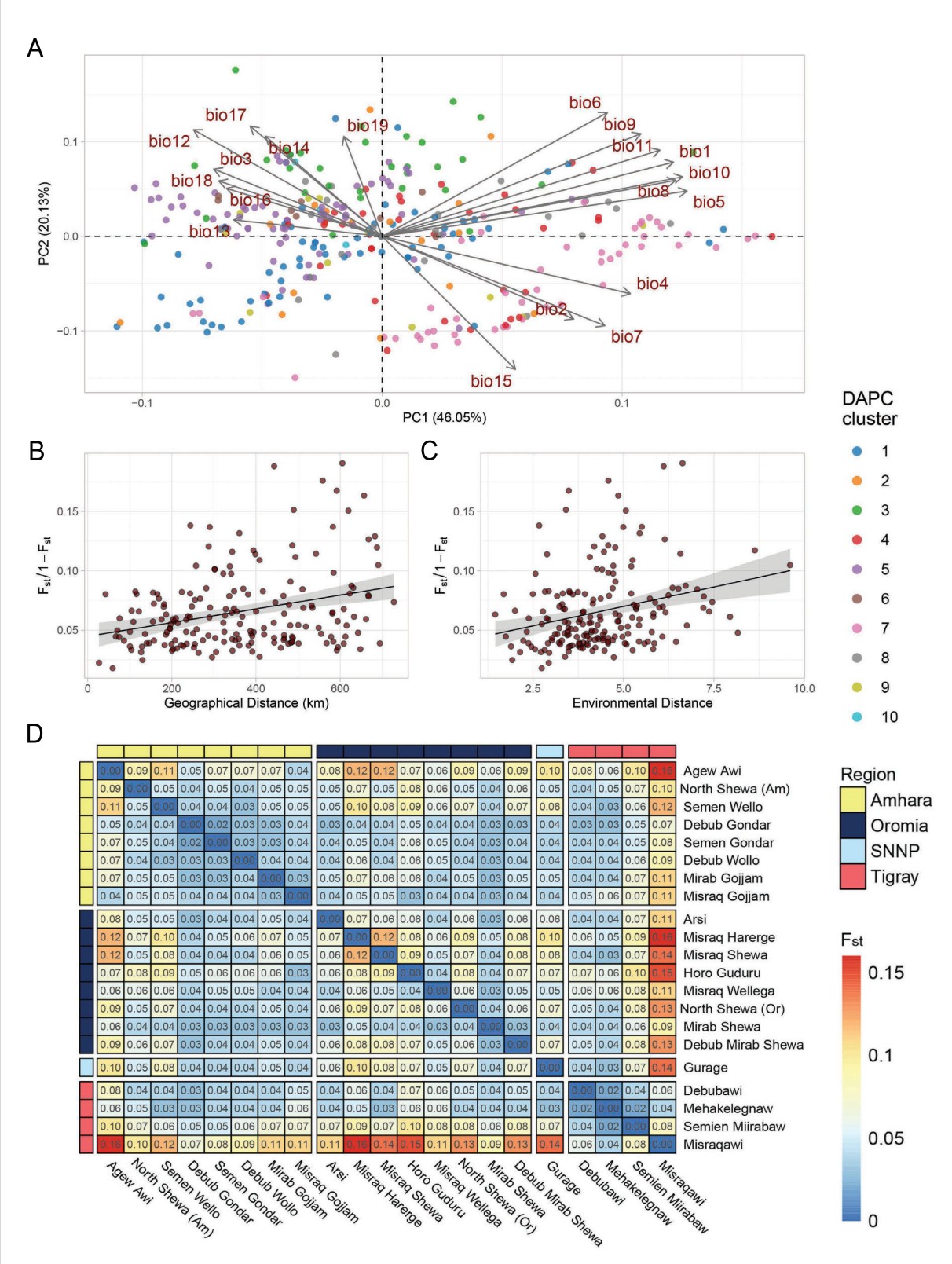

**Figure 2.** Teff diversity on the landscape. (**A**) Principal component analysis of bioclimatic diversity in the Ethiopian Teff Diversity Panel (EtDP). Dots represent teff farmer varieties belonging to genetic clusters, colored according to legend. Vectors represent the scale, verse, and direction of bioclimatic drivers of teff differentiation. (**B**) Linear regression of $F_{st}$ values in relation to geographic distance of accessions in the EtDP. Accessions were grouped by local district of sampling. (**C**) Linear regression of $F_{st}$ values in relation to environmental distance of accessions, also grouped by district of sampling.

*Figure 2 continued on next page*

*Figure 2 continued*

(D) Pairwise $F_{st}$ values between teff accessions grouped by local districts of sampling, as in (C) and (D). Local districts, that is subregional groups, are ordered by administrative regions according to legend. This figure has two figure supplements.

The online version of this article includes the following figure supplement(s) for figure 2:

**Figure supplement 1.** Bioclimatic differences among the 10 discriminant analysis of principal component (DAPC) clusters.

**Figure supplement 2.** Neighbor-joining phylogenetic trees of the Ethiopian Teff Diversity Panel (EtDP) rooted with wild relative accessions *Eragrostis pilosa* and *Eragrostis curvula*.

although challenging to be combined with higher attainable yield. We found that many farmer varieties had a shorter GFP than most improved lines, and that genotype EBI *9551* combined this trait with higher yield and farmers' appreciation (*Figure 2G*).

## Participatory, climatic, and agronomic information to support teff breeding

The data deriving from the transdisciplinary characterization of the EtDP was integrated in a GWAS framework to identify genomic loci associated with agronomic performance, local adaptation, and farmer preferences. These loci could be then used in marker assisted selection, genomic selection, or new breeding technologies to support teff improvement. The GWAS led to the identification of a total of 91 unique quantitative trait nucleotides (QTNs) (*Supplementary file 1E*). Of these, 43 QTNs were associated with bioclimatic variables at sampling sites. Given the low-density genotyping, we did not expect causal loci in the SNP set, yet it is possible to leverage local LD to target candidate genes in the vicinity of associations. Both in the case of trait values and bioclimatic variables, associations may be confounded by underlying LD structure driven by drift processes contributing to teff landraces' differentiation. Teff genetic clusters (*Figure 1*) show different distributions in regards to bioclimatic variables and traits values, including phenology, suggesting that the expression of adaptive traits may be confounded by underlying structure (*Figure 3—figure supplement 2*, *Figure 2—figure supplement 2*). The GWAS analysis was run with varying numbers of genetic covariates, so to optimize model fit in consideration of trait-genotype covariance. Visual assessment of QQ plots was used as a guidance to interpret significant associations, and a stringent threshold was employed so to minimize Type II errors (*Figure 5—figure supplement 2*).

We then focused on LD blocks associated to QTNs to identify homology of predicted teff proteins with protein sequences of *Arabidopsis thaliana* and *Zea mays* (*Supplementary file 1F*). Among others, we identified a locus on chromosome 2A that was associated with grain yield, grain filling rate and OA (lcl|2A-14415768) (Figure 5; *Figure 5—figure supplement 2*). The LD region targeted by this QTN harbors 39 gene models including two 60s ribosomal subunits and several homologs of maize genes with suggestive function in relation to yield determination. *Et_2A_015515*, in this block, is a homolog of a maize serine/threonine protein kinase 3, belonging to a broad class of proteins that was associated with inflorescence development (*McSteen et al., 2007*) and grain yield (*Jia et al., 2020*) in maize. These findings, although preliminary, may support the development of markers for teff improvement as well as directing local and international research toward loci of relevance for teff improvement.

We further explored the EtDP data using a gradient forest (GF) machine-learning algorithm to calculate the contribution of environmental gradients to genomic variation in teff, and to estimate its genomic offset, or vulnerability, under projected climates. We found that the turnover of allele frequencies across the cropping area was best predicted by geographic variables (Moran's eigenvector map variables, MEM), confirming the importance of nonadaptive processes in teff differentiation and the resulting relevance of structure in interpreting associations. MEMs were sided by precipitation indicators, particularly precipitation of the coldest quarter (bio19) and precipitation of the wettest month (bio13; *Figure 4—figure supplement 1*). The GF allowed us to model climate-driven genomic variation patterns across the landscape (*Figure 4A, B*). Approximately a quarter of the SNPs (3,049 in 521 LD blocks) were predictive of the GF model: of these, 176 showed $F_{st}$ values in the 99th percentile of the distribution (*Supplementary file 1G*).

The GF outcome was then linked to the GWAS to identify genomic loci associated to climate, agronomic performance, farmers' preference, and adaptive potential of teff (*Figure 4—figure supplement*

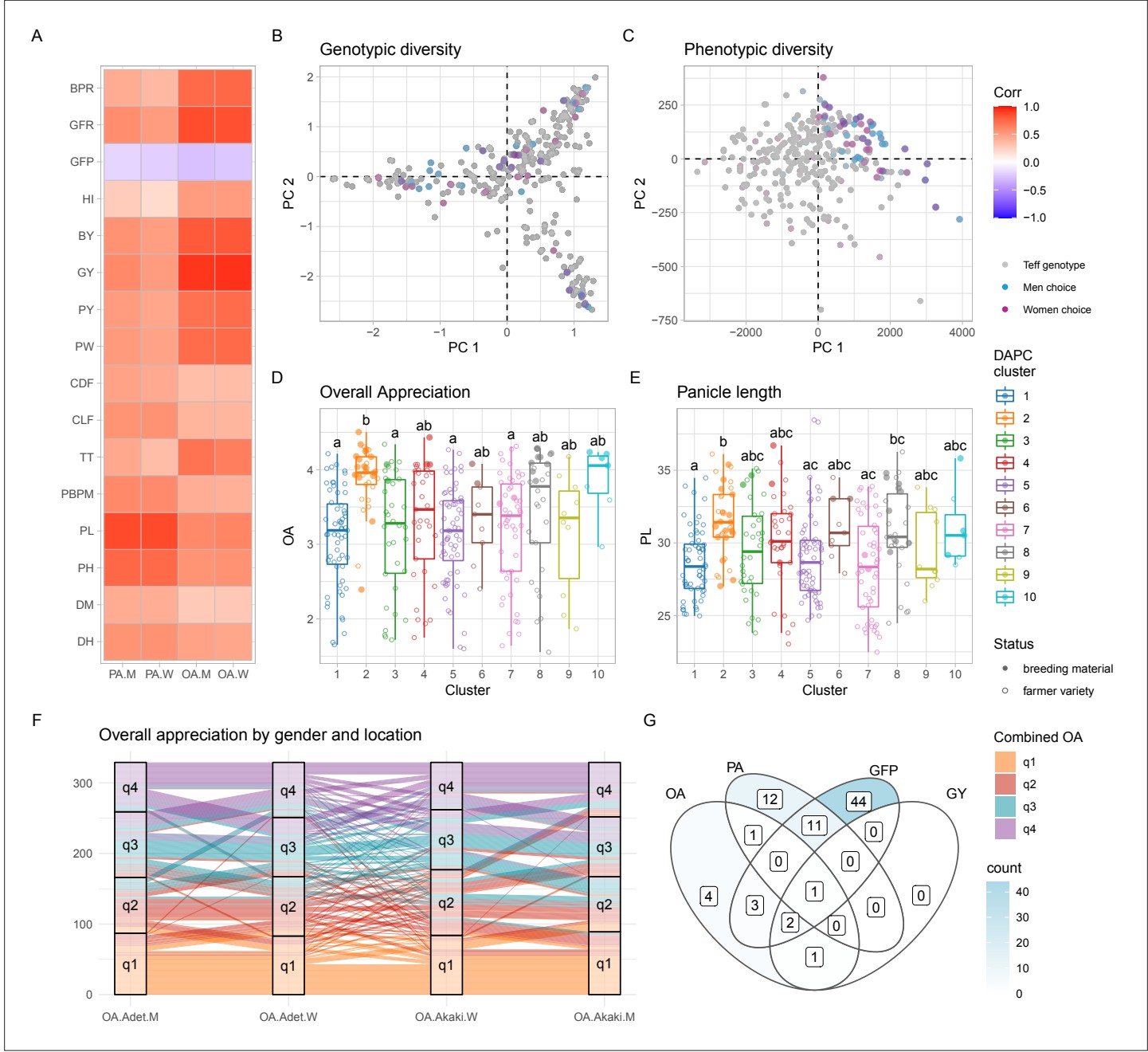

**Figure 3.** Phenotypic diversity in the Ethiopian Teff Diversity Panel (EtDP). (**A**) Pearson's correlations between agronomic traits (*y*-axis) and farmer preference traits, by gender (*x*-axis). Correlation values are expressed in color shades, as indicated in the the legend. (**B, C**) Top ranking genotypes (90th percentile of the OA distribution) selected by men (blue) and women (purple) farmers, overlaid to the principal component analysis (PCA) of the EtDP genetic diversity (B) and phenotypic diversity (**C**). Genotypes that were not selected are reported as gray dots. When men and women select the same genotype, the corresponding point appears in dark violet. Trait distribution across discriminant analysis of principal component (DAPC) clusters, for overall appreciation (**D**) and panicle length (**E**). Farmer varieties are represented by open circles, improved lines are represented by full circles. (**F**) Alluvial plot reporting the consistency of farmers' choice by quartiles of the OA distribution. Each vertical bar represents a combination of location (Adet, Akaki) and gender (M, W). EtDP accessions are ordered on the *y*-axis according to their OA score in each combination. Alluvial flows are colored according to OA quartiles combined across gender and across location according to the legend (q1, q2, q3, and q4). (**G**) Venn diagram reporting farmer varieties having values superior to the 75th percentile of the trait distribution of improved varieties (lower than the 25th percentile in the case of GFP). Each area of the Venn diagram reports the corresponding number of farmer varieties, as in the legend. DH, days to heading; DM, days to maturity; PH, plant height; PL, panicle length; PBPM, number of primary branches per main shoot panicle; TT, total tillers; CLF, first culm length; CDF, first culm diameter; PW, panicle weight; PY, panicle yield; GY, grain yield; BY, biomass yield; HI, harvest index; GFP, grain filling period; GFR, grain filling rate; BPR, biomass production rate; OA, overall appreciation; PA, panicle appreciation. This figure has two figure supplements.

*Figure 3 continued on next page*

*Figure 3 continued*

The online version of this article includes the following figure supplement(s) for figure 3:

**Figure supplement 1.** Correlations between agronomic traits and participatory varietal selection (PVS) traits, by location and gender.

**Figure supplement 2.** Phenotypic differences among the 10 discriminant analysis of principal component (DAPC) clusters.

*2*). We found that phenology QTNs were relevant in supporting GF prediction, although they could not suffice in explaining the geographic distribution of teff diversity (*Figure 4—figure supplement 3*). The importance of phenology in teff adaptation was already reported (*Woldeyohannes et al., 2020*), although it may be confounded by underlying population structure. On chromosome 1A at 32.3 Mb, precipitation of driest month (bio14) and seasonality of precipitations (bio15) identify a QTN (*Figure 4—figure supplement 1*; *Figure 5—figure supplement 2*), whose local LD block harbors 176 gene models, among which four with predicted proteins with high homology with maize and *Arabidopsis* proteins (*Supplementary file 1F*). The product of *Et_1A_007229* is predicted to be homologous to a phosphatidylinositol kinase that is involved in flower development and has been shown to influence floral transition in condition of abiotic stress (*Akhter et al., 2016*). This locus is in the vicinity of a QTN for days to maturity, reinforcing this interpretation. Days to maturity was also associated with a LD block at 20.8 Mb on chromosome 6A (*Figure 4—figure supplement 1*; *Figure 4—figure supplement 2*), in the vicinity of associations for days to heading and for bioclimatic diversity (*Supplementary file 1E*). In this block, nine gene products share homology with *Arabidopsis* and maize (*Supplementary file 1F*). The protein encoded by *Et_6A_046800* is homologous to an ETO1-like protein involved in the regulation of ethylene synthesis in *Arabidopsis* (*Wang et al., 2004*). Ethylene is a key plant hormone that has been shown to be related to spike development and senescence (*Valluru et al., 2017*). These candidate genes have not been validated yet, and their evaluation must be cautious especially for those signals that may be confounded by background genetic structure (*Figure 5—figure supplement 2*). With an increasing refinement of teff gene annotations, corroborated by reverse genetic approaches (*Zhu et al., 2012*), it will be possible to validate genes underlying traits of interest. Teff breeding could then fully benefit from targeted editing (*Lemmon et al., 2018*) to speed up the development of new varieties with improved yield, local adaptation and adherence to local preferences.

The genome-wide teff adaptive potential across the landscape varied in magnitude and distribution according to different predicted climate scenarios for 2070 (*Figure 4—figure supplement 4*). We used these data to compute the genomic-adaptive offset between current and future climate scenarios to identify vulnerable areas (*Figure 4C*, *Figure 4—figure supplement 5*). In all representative concentration pathways (RCPs), the highest offset was predicted in the north-western highlands of the Amhara region, south of lake Tana (*Figure 4C*, *Figure 4—figure supplement 4*). Compared to other regions of the country, we found a decreasing trend of rainfall change in this region across all emission scenarios (*Figure 4—figure supplement 5*; *Figure 4—figure supplement 6*). In this area, hot nights are projected to increase more quickly than hot

**Table 1.** Heritabilities ($H^2$) for farmers' participatory variety selection traits.

$H^2$ values are given for each trait, type, and location combination. PA, panicle appreciation; OA, overall appreciation; M, men; W, women; ALL, measures combined by either type or location.

| Trait | Type | Location | $H^2$ |
|-------|------|----------|-------|
| PA | ALL | ALL | 0.48 |
| PA | ALL | Adet | 0.50 |
| PA | ALL | Akaki | 0.65 |
| PA | M | ALL | 0.43 |
| PA | W | ALL | 0.45 |
| PA | M | Adet | 0.35 |
| PA | W | Adet | 0.40 |
| PA | M | Akaki | 0.25 |
| PA | W | Akaki | 0.34 |
| OA | ALL | ALL | 0.81 |
| OA | ALL | Adet | 0.71 |
| OA | ALL | Akaki | 0.87 |
| OA | M | ALL | 0.74 |
| OA | W | ALL | 0.74 |
| OA | M | Adet | 0.53 |
| OA | W | Adet | 0.66 |
| OA | M | Akaki | 0.88 |
| OA | W | Akaki | 0.62 |

**Table 2.** Heritabilities ($H^2$) for agronomic traits. $H^2$ values are given for each trait, and location combination. DH, days to heading; DM, days to maturity; PH, plant height; PL, panicle length; PBPM, number of primary branches per main shoot panicle; TT, total tillers; CLF, first culm length; CDF, first culm diameter; PW, panicle weight; PY, panicle yield; GY, grain yield; BY, biomass yield; HI, harvest index; GFP, grain filling period; GFR, grain filling rate; BPR, biomass production rate; ALL, measures combined by location.

| Trait | Location | $H^2$ |
|---|---|---|
| DH | ALL | 0.99 |
| DH | Adet | 0.96 |
| DH | Akaki | 0.97 |
| DM | ALL | 0.98 |
| DM | Adet | 0.90 |
| DM | Akaki | 0.89 |
| PH | ALL | 0.16 |
| PH | Adet | 0.92 |
| PH | Akaki | 0.90 |
| PL | ALL | 0.37 |
| PL | Adet | 0.88 |
| PL | Akaki | 0.81 |
| PBPM | ALL | 0.64 |
| PBPM | Adet | 0.83 |
| PBPM | Akaki | 0.83 |
| TT | ALL | 0.25 |
| TT | Adet | 0.65 |
| TT | Akaki | 0.60 |
| CLF | ALL | 0.13 |
| CLF | Adet | 0.88 |
| CLF | Akaki | 0.85 |
| CDF | ALL | 0.15 |
| CDF | Adet | 0.95 |
| CDF | Akaki | 0.96 |
| PW | ALL | 0.27 |
| PW | Adet | 0.90 |
| PW | Akaki | 0.88 |
| PY | ALL | 0.25 |
| PY | Adet | 0.92 |
| PY | Akaki | 0.91 |

*Table 2 continued on next page*

*Table 2 continued*

| Trait | Location | $H^2$ |
|---|---|---|
| GY | ALL | 0.42 |
| GY | Adet | 0.92 |
| GY | Akaki | 0.93 |
| BY | ALL | 0.34 |
| BY | Adet | 0.84 |
| BY | Akaki | 0.80 |
| HI | ALL | 0.78 |
| HI | Adet | 0.68 |
| HI | Akaki | 0.76 |
| GFP | ALL | 0.96 |
| GFP | Adet | 0.79 |
| GFP | Akaki | 0.83 |
| GFR | ALL | 0.52 |
| GFR | Adet | 0.90 |
| GFR | Akaki | 0.92 |
| BPR | ALL | 0.32 |
| BPR | Adet | 0.80 |
| BPR | Akaki | 0.77 |

days, with the most marked increases expected to be experienced in the July, August, September season (*Figure 4—figure supplement 7*). Decreasing trends of rainfall during the main growing season are predicted in all projected scenarios, suggesting that seasonality might critically impact teff development stages (*Figure 4—figure supplement 6*). In light of these results, a valid adaptation strategy could be the assisted migration of teff genetic backgrounds from areas of different vulnerability (*Rhoné et al., 2020*). However, crop migration and varietal replacement strategies will need to take into account ecological and socioeconomic factors, including the impacts on existing ecosystems and on farmers' adoption of migrated varieties (*Sloat et al., 2020*).

## Conclusion

A comprehensive interpretation of crop performance is key to a sustainable intensification that embraces cultural and agricultural diversity of cropping systems. While significant successes and even a plateau might have been reached in optimal growing environments where most common crops are cultivated, there is ample opportunity to enhance productivity in marginal growing environments (*Godfray et al., 2010*).

**Table 3.** Plackett–Luce estimates from farmer's overall appreciation (OA) of genotypes associated with genotypes' agronomic metrics, DM, days to maturity; PH, plant height; PW, panicle weight; GY, grain yield; BY, biomass yield; GFP, grain filling period.

The rankings were analyzed for the whole group (All) and in subsets among gender to assess differences in traits linkages within men and women farmers. * $0.05 < p < 0.01$, ** $0.01 < p < 0.001$, *** $p < 0.001$.

| Group | | Estimate | Std. error | z value | Pr(>\|z\|) | |
|---|---|---|---|---|---|---|
| **All** | (Intercept) | −7.09 | – | – | – | |
| | GY | 0.000764 | 0.000135 | 5.66 | 1.51E−08 | *** |
| | DM | 0.00625 | 0.00402 | 1.56 | 0.12 | |
| | PH | −0.0123 | 0.00275 | −4.46 | 8.17E−06 | *** |
| | PW | −0.0557 | 0.0716 | −0.777 | 0.437 | |
| | BY | −3.6E−05 | 2.72E−05 | −1.32 | 0.186 | |
| | GFP | 0.011 | 0.00484 | 2.27 | 0.0233 | * |
| **Men** | (Intercept) | −7.04 | – | – | – | |
| | GY | 0.000765 | 0.000175 | 4.36 | 1.29E−05 | *** |
| | DM | 0.00542 | 0.00531 | 1.02 | 0.308 | |
| | PH | −0.0114 | 0.00361 | −3.17 | 0.00155 | ** |
| | PW | −0.0818 | 0.0932 | −0.878 | 0.38 | |
| | BY | −2E−05 | 0.000036 | −0.562 | 0.574 | |
| | GFP | 0.00919 | 0.00637 | 1.44 | 0.149 | |
| **Women** | (Intercept) | 15.6 | – | – | – | |
| | GY | −0.00084 | 0.00028 | −3.01 | 0.00262 | ** |
| | DM | −0.186 | 0.0101 | −18.5 | <2e−16 | *** |
| | PH | −0.139 | 0.00661 | −20.9 | <2e−16 | *** |
| | PW | 4.4 | 0.195 | 22.6 | <2e−16 | *** |
| | BY | 0.000346 | 5.43E−05 | 6.38 | 1.74E−10 | *** |
| | GFP | −0.136 | 0.0104 | −13.1 | <2e−16 | *** |

The success of crop varieties is not only determined by yield performance, but also by adaptation to local environments and cultural needs (*Weltzien et al., 2019*). The integration of genomic, climatic, and phenotyping diversity in a participatory framework may help tailor varietal development for local adaptation. The involvement of farmers in varietal evaluation is increasingly utilized in a quantitative framework to guide breeding choices in combination with genomic data (*Annicchiarico et al., 2019*; *de Sousa et al., 2021*).

Transdisciplinary methods may support the integration of smallholder farmers in modern breeding and agricultural value chains. Modern data-driven research may then efficiently harness the genetic diversity generated in farmers' fields to project NUS, such as teff, into modern breeding. Genebank genomics may be systematically used to map its large agrobiodiversity (*Woldeyohannes et al., 2020*), fully disclosing the opportunity to use next generation breeding technologies (*Varshney et al., 2021*) and large-scale genomic selection (*Poland, 2015*). Multilocation trials are needed to capture the range of G × E interactions that influence agronomic performance of teff, which we could only partially characterize here. Decentralized varietal evaluation approaches can be used to scale up the testing of teff genetic resources (*de Sousa et al., 2021*; *van Etten et al., 2019a*) to better capture G × E, produce new varieties with higher local adaptation, and resulting in higher farmers' varietal adoption. An extensive socioeconomic characterization of teff cropping, including extensive farmers interviews (*Labeyrie et al., 2016*), may further unveil the dynamics of teff seed exchange and associated

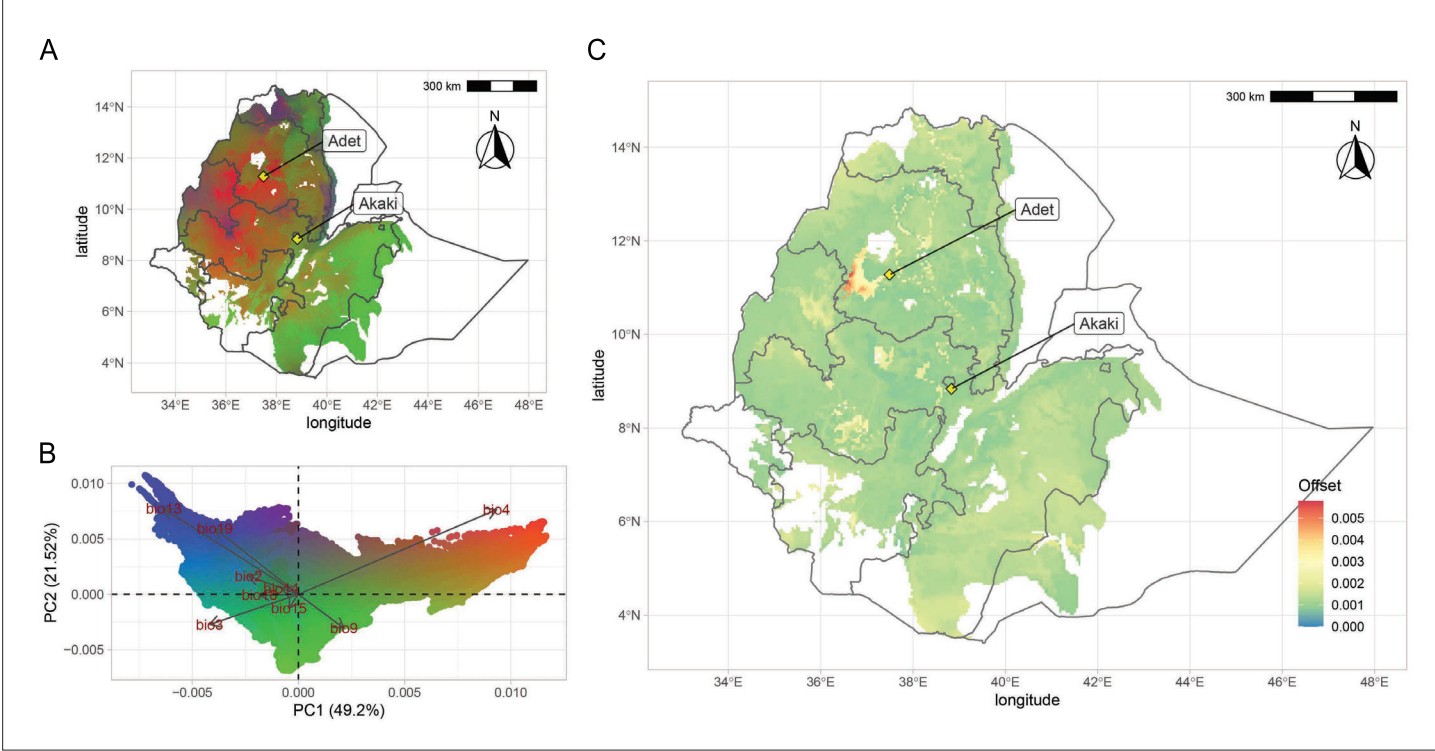

**Figure 4.** Teff genomic offset. (**A**) Geographic distribution of climate-driven allelic variation under current climates across the teff cropping area, with colors representing the three principal component (PC) dimensions reported in panel (**B**). (**C**) Genomic vulnerability across the teff cropping area based on the RCP8.5 climate projections. The color scale indicates the magnitude of the mismatch between current and projected climate-driven turnover in allele frequencies according to legend. Phenotyping locations are shown with yellow diamonds. This figure has seven figure supplements.

The online version of this article includes the following figure supplement(s) for figure 4:

**Figure supplement 1.** Ranked accuracy and importance of bioclimatic (**A**) and geographic (Moran's eigenvector map [MEM]).

**Figure supplement 2.** Linkage disequilibrium (LD) blocks distribution in relation to $F_{st}$ (y-axis) and GF $r^2$ values (x-axis).

**Figure supplement 3.** Geographic distribution of climate-driven allelic variation under four representative concentration pathways: (**A**) RCP2.6, (**B**) RCP4.5, (**C**) RCP6.0, and (**D**) RCP8.5.

**Figure supplement 4.** Genomic vulnerabilities in the teff cropping area based on projections for four representative concentration pathways: (**A**) RCP2.6, (**B**) RCP4.5, (**C**) RCP6.0, and (**D**) RCP8.5.

**Figure supplement 5.** Projected change in Ethiopian climate for 2070s under RCP8.5 compared to 1986–2005.

**Figure supplement 6.** Mean change in monthly rainfall compared to the reference period in the south Lake Tana (36.5–37.75 east,10.7–12 north) by 2070 under RCP2.6 (low emissions), RCP4.5 (medium-low emission), RCP6.0 (medium-high emission), and RCP8.5 (high emission) scenarios.

**Figure supplement 7.** Mean change in monthly temperature compared to the reference period in the south Lake Tana (36.5–37.75 east, 10.7–12 north) by 2070 under RCP2.6 (low emissions), RCP4.5 (medium-low emission), RCP6.0 (medium-high emission), and RCP8.5 (high emission) scenarios.

flow of information and knowledge related to local cropping (*Occelli et al., 2021*). Public sector breeding beyond NUS may further enhance the combination of data-driven research with participatory approaches to improve customer and product profiling to achieve social impact as cost effective as possible.

The Intergovernmental Panel on Climate Change (IPCC) reports indicate that East Africa will experience an increase in aridity and agricultural droughts, with a substantially higher frequency of hot days and nights (*IPCC, 2017*). Temperature increases are also expected to result in more intense heat waves and higher evapotranspiration rates, which, coupled with the altered rainfall patterns, may affect agricultural productivity. Enhancing NUS farmer varieties offers promising opportunities to tackle food insecurity resulting from climate change in smallholder farming settings and beyond. NUS have enormous untapped potential for improvement that is hampered by lack of tools and knowledge (*Yerima and Achigan-Dako, 2021*), but our analyses show that their characterization is at hand.

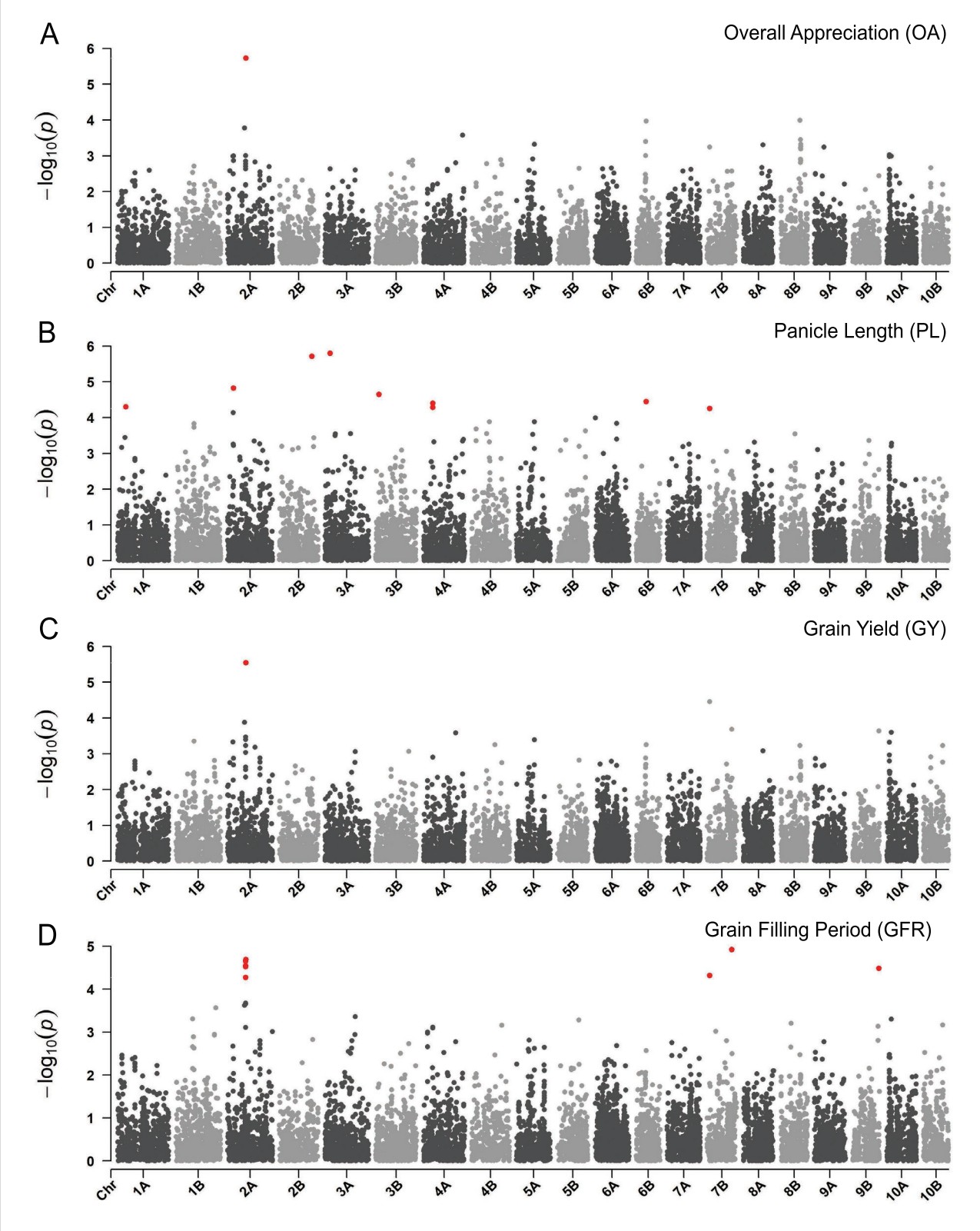

**Figure 5.** Manhattan plots reporting the genome-wide association study (GWAS) result for. (**A**) Overall appreciation, (**B**), panicle length, (**C**) grain yield, and (**D**) grain filling period. On the x-axis, the genomic position of markers. The y-axis reports the strength of the association signal. Single-nucleotide polymorphisms (SNPs) are ordered by physical position and grouped by chromosome. Quantitative trait nucleotides (QTNs), for example SNPs

*Figure 5 continued on next page*

*Figure 5 continued*

surpassing a threshold based on a false discovery rate of 0.05, are highlighted in red. A strong signal on chromosome 2A matches across participatory varietal selection (PVS) and metric traits. This figure has two supplements.

The online version of this article includes the following figure supplement(s) for figure 5:

**Figure supplement 1.** Manhattan plots reporting the genome-wide association study (GWAS) result for (**A**) precipitation of the driest month, (**B**) precipitation seasonality and (**C**) days to maturity. On the *x*-axis, the genomic position of markers.

**Figure supplement 2.** Quantile–quantile plots for the genome-wide association study (GWAS) scans reported in *Figure 5A, C and D*.

As NUS proceed toward mainstream breeding, the collaborative effort of scientists, breeders, and farmers will unlock their full potential for sustainable intensification of farming systems.

# Materials and methods
## Plant materials

The EtDP used in this study was derived from a larger teff collection of 3850 accessions held at the Ethiopian Biodiversity Institute (EBI; Addis Abeba, Ethiopia), which represents the world's largest active teff collection, and was amplified and characterized by *Woldeyohannes et al., 2020*. Criteria for selecting the EtDP from the larger EBI collection were the following: (1) visible morphological variation for panicle traits, (2) geographical and agroecological representativeness, (3) apparent grain yield potential, (4) presence of different maturity groups, and (5) presence of associated traditional names of specific landraces. The EtDP includes 321 farmer varieties, sided by all 45 *E. teff* improved varieties released since the beginning of teff breeding program in Ethiopia and until the assembly of the EtDP. Improved varieties were obtained by Ethiopian agricultural research centers. Seven accessions of teff wild relatives *E. pilosa* and *E. curvula* were also included in the collection (*Supplementary file 1A*). Landraces were purified selecting and reproducing one single panicle representative of each accession. The ITPGRFA defines a crop variety as '[…] defined by the reproducible expression of its distinguishing and other genetic characteristics' (*Ho, 2011*). For the scope of this paper, we define farmer varieties as uniform genotypes derived from the purification of *ex situ* accessions (i.e., landraces) collected from local farmers. Farmer varieties are therefore a proxy of landraces originally collected in farmer fields and are discussed as such.

## Sequencing and variant calling

Seeds of the EtDP were germinated in pots at the EBI in 2018, and at least three seedlings were harvested and pooled per accession. Genomic DNA was extracted from pooled seedlings at the EBI laboratories using the GenElute Plant Genomic DNA Miniprep Kit (Sigma-Aldrich, St. Louis, MO, USA) following the manufacturer's instructions. DNA quality was checked by electrophoresis on 1% agarose gel and using a NanoDrop ND-1000 spectrophotometer and sent to IGATech (Udine, Italy) for sequencing. Genomic libraries were produced using *SphI* and *MboI* restriction enzymes in a custom protocol for the production of double digestion restriction site-associated DNA markers (ddRAD; *Peterson et al., 2012*). In short, ddRAD is based on a double restriction of the target DNA, typically using a rare cutter and a frequent cutter, followed by sequencing of restriction fragments to reduce the complexity of the target genome. ddRAD libraries were sequenced with V4 chemistry on Illumina HiSeq2500 sequencer (Illumina, San Diego, CA) with 125 cycles in a paired-end mode. Reads were demultiplexed using the *process_radtags* utility included in Stacks v2.0 (*Catchen et al., 2013*) and analyzed for quality control with the FastQC tool (v.0.11.5). High-quality paired-end reads of each individual were mapped against the *E. tef* reference genome (version 3, available from CoGe under ID 50954; *VanBuren et al., 2020*) with BWA (Burrows-Weeler-Aligner v.0.7.12) using the MEM algorithm with standard parameters (*Li, 2013*). Alignments were sorted and indexed with PicardTools (http://broadinstitute.github.io/picard/) and samtools (*Li et al., 2009*). Single-nucleotide variants were identified with GATK (*McKenna et al., 2010*) HaplotypeCaller algorithm (version 4.2.0), run in per-sample mode followed by a joint genotyping step completed by GenotypeVCFs tool. Raw variants were filtered out using the VariantFiltration and SelectVariants GATK functions with the following criteria: monomorphic or multiallelic sites, QUAL <30; QD <2.0; MQ <40.0; AF <0.01; DP <580; *SNP clusters* defined as three or more variants located within windows of 5 bp. For each accession, SNPs with a

total read count of <3 were set to NA. Variants were discarded if located on unanchored contigs, InDels, missing data >20%, heterozygosity >15%.

## Bioclimatic characterization

GPS coordinates of EtDP teff landraces were derived from EBI passport data and projected onto the map of Ethiopia using R/raster (*Hijmans and Etten, 2012*). Teff landraces were projected onto the agroecological zones map of Ethiopia provided by the Ethiopian Institute of Agricultural Research (EIAR; *MoA, 2000*), which subdivides Ethiopia into different zones according to altitudinal ranges and temperature and rainfall patterns. Altitudes were assigned to each landrace based on GPS coordinates, using the CGIAR SRTM database at 90 m resolution (*Reuter et al., 2007*). Current climate data (1970–2000 averages) relative to teff landraces' sampling sites were retrieved from the WorldClim 2 database of global interpolated climate data (*Fick and Hijmans, 2017*) at the highest available spatial resolution of 2.5′ (approximately 4 × 4 km). Collinearity among historical bioclimatic variables was previously checked with the *ensemble.VIF()* function in R/BiodiversityR (*Kindt and Coe, 2005*). Only variables with a variation inflation factor (VIF) below 10 were retained, namely bio2, bio3, bio4, bio9, bio13, bio14, bio15, bio18, and bio19. The Hadley Centre Global Environmental Model 2-Earth System (HadGEM2-ES; *Jones et al., 2011*) under the fifth phase of the Coupled Model Intercomparison Project (CMIP5) protocols simulations was used to retrieve future climate scenarios at the following RCPs (RCP2.6, RCP4.5, RCP6.0, and RCP8.5).

## Phenotyping and participatory variety selection

The EtDP was phenotyped in common garden experiments in two high-potential teff growing locations in Ethiopia, Adet (Amhara, 11°16′32″ N, 37°29′30″ E) and Akaki (Oromia, 8°50′07.6″ N, 38°49′58.3″ E), under rainfed conditions during the main cropping season of 2018 (July–November). Adet is the main experimental site of the Amhara Regional Agricultural Research Institute, at an altitude of 2240 meter above sea level. Soil type is vertisol, climate is moist cool, and average annual rainfall is 1250 mm. Akaki is a subsite of the Debre Zeit Agricultural Research Center, with an altitude of 2200 meter above sea level. Also in this case, soil type is vertisol, climate is moist cool and average annual rainfall is 1055 mm. Accessions were planted in two replications per site using an alpha lattice design, in plots consisting of three rows of 1 m in length and 0.2 m interrow distance. Three phenological traits, days to 50% heading (DH), days to 90% maturity (DM), and GFP were recorded on whole plots in each environment. Morphology and agronomic traits were recorded from five randomly selected teff plants per plot: plant height (PH, cm), panicle length (PL, in cm), number of primary branches per main shoot panicle (PBPM), number of total tillers (TT), first culm length (CLF, in cm), first culm internode diameter (CDF, in mm), panicle weight (PW, in g), panicle yield (PY, in g), grain yield (GY, ton/ha), biomass yield (BY, ton/ha), harvest index (HI), grain yield filling rate (GFR, kg/ha/day), and biomass production rate (BPR, kg/ha/days). Qualitative data were sourced from the characterization performed by *Woldeyohannes et al., 2020*.

A PVS was conducted in the two locations, involving 35 experienced teff farmers: 15 men and 10 women in Adet, five men and five women in Akaki. PVS was conducted close to physiological maturity in each location so to maximize variation between plots. Participating farmers were selected with the help of agricultural officers. They all had experience on teff production and were recognized as a local agricultural expert in teff cultivation. Being local farmers, they spoke different languages (Amharic and Oromo) and local interpreters were employed. Farmers were engaged in focus group discussions prior to the PVS to discuss most relevant traits in teff cultivation and to attend training on how to perform the PVS. During the evaluation, farmers were divided into gender-homogeneous groups with five people each. Groups were conducted across the field from random entry points and asked to evaluate two traits: PA and OA. PVS traits were given on a Likert scale from 1 (poor) to 5 (excellent), in a way answering a question in the form of: 'how much do you like the [panicle appearance/overall appearance] of this plot from one to five?'. Farmers provided their scores simultaneously so that within-group scoring bias was reduced. Each farmers' score was recorded individually. We used Plackett–Luce model (*Turner et al., 2019*) to analyze farmers' OA. Agronomic traits from tested genotypes were linked to farmer's OA with a linear model using an Alternating Directions Method of Multipliers (ADMM) algorithm proposed by *Yıldız et al., 2020*.

## Phenotypic data analyses

We used an analysis of variance (ANOVA) to describe trait differences conditional to teff genetic groups, locations, and, in the case of PVS, to gender with a procedure similar to that used in *Mokuwa et al., 2013*. Best linear unbiased predictions (BLUPs) of agronomic and PVS traits were computed with R/ASReml (*Gilmour et al., 2014*). BLUPs for agronomic traits were derived from the general model in *Equation S1*:

$$y_{ik} = \mu + g_i + l_k + gl_{ik} + e \tag{S1}$$

where the observed phenotypic value is $y_{ik}$, $\mu$ is the overall mean of the population, $g_i$ is the random effect for the *i*th genotype $g$, $l_k$ is the fixed effect for the *k*th location, $gl_{ik}$ is the random effect interaction between genotype and location, and $e$ is the error. For calculation of BLUPs with a single location, the data were subset by location and the model in *Equation (S1)* was simplified accordingly. Broad-sense heritability ($H^2$) of agronomic traits was derived from the variance component estimates deriving from *Equation (S1)* as follows:

$$H^2 = \frac{\sigma_g}{\left(\sigma_g + \frac{\sigma_{gl}}{n_{loc}} + \frac{\sigma_e}{n_{rep}*n_{loc}}\right)} \tag{S2}$$

In *Equation (S2)*, $\sigma_g$ is the variance component of genotypes, $\sigma_{gl}$ is the genotype by location variance, and $\sigma_e$ is the error variance. $n_{loc}n_{rep}$ are the number of locations and replications, respectively. For calculation of $H^2$ within locations (i.e., repeatability), *Equation (S2)* was simplified accordingly.

The derivation of PVS BLUPs and $H^2$ was like that used for agronomic traits except for the fact that gender of farmers was considered. BLUPs for PVS were obtained from the model in *Equation (S3)*:

$$y_{ikm} = \mu + g_i + l_k + p_m + gl_{ik} + gp_{im} + pl_{mk} + e \tag{S3}$$

where $y_{ikm}$ is the observed PVS score, and $\mu$, $g_i$, $l_k$, and $gl_{ik}$ are as in *Equation (S1)* and $p_m$ is the random effect for farmer gender. Accordingly, $gp_{im}$ is the random effect of the interaction between genotype and gender and $pl_{mk}$ is the random interaction between gender *m* and the *k*th location. For calculation of BLUPs specific for gender, location, and gender by locations, *Equation (S3)* was simplified accordingly. $H^2$ for PVS traits was derived from the following formula:

$$H^2 = \frac{\sigma_g}{\left(\sigma_g + \frac{\sigma_{gl}}{n_{loc}} + \frac{\sigma_{gm}}{n_{gender}} + \frac{\sigma_e}{n_{rep}*n_{loc}*n_{gender}*n_{farmer}}\right)} \tag{S4}$$

In *Equation (S4)*, $\sigma_g$ is the variance component of genotypes, $\sigma_{gl}$ is the genotype by location variance, $\sigma_{gm}$ is the genotype by gender variance, and $\sigma_e$ is the error variance. $n_{loc}$, $n_{gender}$, and $n_{rep}$ are the number of locations, genders, and replications, respectively. For calculation of $H^2$ (i.e., repeatability) by gender and by location, *Equation (S4)* was simplified accordingly. The 90th percentile of the OA distribution was considered to identify top ranking accessions for men and women. Farmer varieties were benchmarked with the fourth quartile of the distribution of improved lines for all scored traits.

## Genetic diversity

Phylogenetic relationships in the EtDP were assessed on a pruned set of SNP markers with MAF >0.05. Pruning was performed with the PLINK (*Purcell et al., 2007*) *indep-pairwise* function on a 100 SNPs window moving in 10 SNP steps with an LD $r^2$ threshold of 0.3. Pairwise identity by descent (IBS) was calculated by PLINK and visualized with custom scripts in R (*R Development Core Team, 2018*). PCA and DAPCs were performed with R/adegenet (*Jombart, 2008*). The optimal number of clusters (*K*) for the DAPC was identified using *find.cluster()*. Bayesian information criterion (BIC) statistics were computed at increasing values of *K* to measure the goodness of fit at each *K* using 365 PCs and default settings. ADMIXTURE (*Alexander and Lange, 2011*) was run testing 2–25 K clusters using the default termination criterion. Each iteration was run using different random seeds, and parameter standard errors were estimated using 2000 bootstrap replicates. A fivefold cross-validation procedure was used to identify the most likely value of K. The correlation of the residual difference between the true genotypes and the genotypes predicted by the model was estimated using EvalAdmix (*Garcia-Erill and Albrechtsen, 2020*). A neighbor-joining (NJ) tree was developed computing genetic distances using the Tajima–Nei method (*Tajima and Nei, 1984*) and performing 500 bootstrap resampling,

using MEGA X (*Kumar et al., 2018*). Different NJ tree visualizations were produced using R/ggtree (*Yu et al., 2017*). Genotypic data of putative teff wild relatives *E. curvula* and *E. pilosa* were integrated in the NJ phylogeny. A set of putative SNPs shared between wild relatives and cultivated teff was derived as described for the EtDP.

LD analyses were performed on SNPs with MAF >0.05. Average pairwise $r^2$ for all markers within a window of ±5 Mb was estimated using R/LDheatmap (*Shin et al., 2007*). The LD was plotted against physical positions, averaging pairwise $r^2$ values for each chromosome over sliding window considering portions equal to 5% of each chromosome's physical length. LD decay was then estimated for each of the chromosomes (*Hill and Weir, 1988*) using a threshold of $r^2 = 0.3$. Haplotype blocks were estimated using the PLINK-blocks function with default settings and following the interpretation of *Gabriel et al., 2002*.

## Climatic diversity and GF

Agroecological and bioclimatic variation analyses were performed on georeferenced materials of the EtDP. The distribution of the DAPC genetic clusters across agroecological zones was mapped via R/raster. After aggregating teff georeferenced accessions in Ethiopian administrative regions at the second level (districts), pairwise $F_{st}$ (*Weir and Cockerham, 1984*) was calculated across all SNP markers for all areas accounting at least five individuals. Centroid coordinates of the accessions within each district were used to estimate geographic distances, while environmental distances were calculated by averaging the value of noncorrelated historical bioclimatic variables and altitude. A measure of environmental distance between each accession was thus calculated as pairwise Euclidean differences between locations. A Mantel test with a Monte Carlo method (9999 replications) was implemented in R/ade4 (*Dray and Dufour, 2007*) to check associations between linearized $F_{st}$ ($F_{st}/1 - F_{st}$) and geographic and environmental distances.

The teff cropping area was defined by the union of all polygons representing agroecological zones in which at least two teff landraces were sampled. Significant associations between genetic clusters and agroecological zones and administrative regions were assessed using Pearson's chi-squared test of independence. Pairwise Wilcoxon rank sum test was used to test the significance ($p < 0.05$) of differences in bioclimatic variables among DAPC clusters. A GF machine-learning approach implemented in R/gradientForest (*Ellis et al., 2012*; *Fitzpatrick and Keller, 2015*) was used to map the turnover in allele frequencies using nonlinear functions of environmental gradients with historical and projected climates. The GF was developed using historical noncollinear bioclimatic variables and MEM variables representing climatic and geographic diversity in the sample, respectively. MEM variables were derived from geographic coordinates at sampling locations of the landraces in the EtDP (*Dray et al., 2006*; *Griffith and Peres-Neto, 2006*) and were calculated with (*dbmem*) in R/adespatial (*Stéphane Dray et al., 2021*). A function was built for each response variable (SNPs) using 500 regression trees. An aggregate function was created for all SNPs, whereas the bioclimatic variables and MEMs were used as predictors. The model was then run to predict teff genetic–geographic–climatic distribution on the teff cultivation range in Ethiopia. The GF model was also run using and projected climate data under different RCP scenarios.

Climate projections for areas of interest were analyzed to assess trends in rainfall and temperature. The 12 models best performing in the East Africa region according to IPCC (*IPCC, 2017*) were used to develop and ensemble projection of rainfall and temperature indices with Climate Data Operators (CDO) (*Schulzweida, 2017*) and custom R scripts. Projected data were compared with historical data to derive indices change in the interannual variability for the regions of interest.

## Genome-wide association studies

QTNs were mapped in a GWAS. GWAS was performed with R/rMVP (*Yin et al., 2019*) using the Fixed and random model Circulating Probability Unification (FarmCPU) method (*Liu et al., 2016*) that incorporates corrections for population cryptic relatedness (Kinship). The first 10 genetic PCs were used as covariates to account for population structure. The Kinship was estimated using the method implemented by *VanRaden, 2008*. Both kinship and PCA were calculated using the subset of LD-pruned markers used for population genetics analysis. GWAS was run on bioclimatic variables, agronomic traits, and PVS traits. After the first round of mapping with 10 PCs, individual QQ plot was visually surveyed for inflation in the p value distributions, as these could be caused by suboptimal

correction for population structure. When inflation was detected, the corresponding GWAS scan was run again using 25 genetic PCs as covariates. QTN was called when association surpassed a multiple testing correction with false discovery rate of 5% using $R/q$ value (**Storey et al., 2021**). QTNs were assigned to the previously defined haplotype blocks. Blocks were extended by the chromosome-specific LD decay distance upstream and downstream and used as windows to search for candidate genes. The LD blocks thus obtained were combined with $F_{st}$ and GF results to identify intersections across methods.

Teff gene annotations were retrieved from CoGe under id50954 (**VanBuren et al., 2020**). Nucleotide sequences of putative candidate genes were translated into the corresponding proteins and used as queries against Araport11 (**Cheng et al., 2017**) and the Maize reference proteome, available from UniProt (https://www.uniprot.org/) under the ID UP000007305. $E$ value of $10^{-20}$ and percentage of identity of 50% were used as threshold to retain blast hits on *Arabidopsis* and maize.

## Acknowledgements

We wish to thank the men and women farmers who took part to the research, evaluating hundreds of genetic materials with great enthusiasm and competence in Adet and Akaki. These are their names in alphabetical order: Abebech, Adisse Abera, Adugawu Kere, Alebachew, Angoachi Fente, Askale, Ayana, Belachew Adimasu, Belay, Birtukan Ayele, Birhanu, Genanaw Dejene, Getasil, Habtamu Belay, Mame Seyum, Mamo, Misganaw Dagnaw, Regassa, Sefiwu Kassahun, Shashe, Tenagne Wubet, Tesfa Kefyalew, Tihune Dires, Tirusew, Tsehay Desie, Workinesh, Workineh Tsega, Wubet, Wubitu, Yechale Ayalneh, Yekoye, Yeniguse Worku, Yeniguse Wuletaw, Zebu Kebede, and Zina Yitay. We thank the Ethiopian Biodiversity Institute (EBI) for providing seed material and laboratory space to perform the DNA extraction in Ethiopia. Many thanks to Dr. Eleni Shiferaw, Dr. Basazen Fantahun Lakew, and Dr. Yosef Gebrehawairat Kidane for coordinating local support at EBI. We are grateful to the reviewers for constructive criticism and help in improving the manuscript. Funding was provided by the Doctoral School in Agrobiodiversity at Scuola Superiore Sant'Anna.

## Additional information

### Funding

| Funder | Grant reference number | Author |
| --- | --- | --- |
| Scuola Superiore Sant'Anna | | Aemiro Bezabih Woldeyohannes |

The funders had no role in study design, data collection, and interpretation, or the decision to submit the work for publication.

### Author contributions

Aemiro Bezabih Woldeyohannes, Resources, Data curation, Formal analysis, Investigation, Writing – original draft; Sessen Daniel Iohannes, Resources, Data curation, Software, Formal analysis, Visualization, Writing – original draft; Mara Miculan, Resources, Data curation, Software, Formal analysis, Investigation, Writing – review and editing; Leonardo Caproni, Formal analysis, Investigation, Visualization, Methodology, Writing – review and editing; Jemal Seid Ahmed, Software, Formal analysis, Validation, Writing – review and editing; Kauê de Sousa, Data curation, Investigation, Visualization, Writing – review and editing; Ermias Abate Desta, Mario Enrico Pè, Conceptualization, Supervision, Project administration, Writing – review and editing; Carlo Fadda, Conceptualization, Supervision, Funding acquisition, Project administration, Writing – review and editing; Matteo Dell'Acqua, Conceptualization, Data curation, Supervision, Validation, Visualization, Methodology, Writing – original draft, Project administration

### Author ORCIDs

Sessen Daniel Iohannes (iD) http://orcid.org/0000-0002-7831-8997
Mara Miculan (iD) http://orcid.org/0000-0002-9884-5727
Leonardo Caproni (iD) http://orcid.org/0000-0002-7129-8575

Kauê de Sousa [ORCID] http://orcid.org/0000-0002-7571-7845
Matteo Dell'Acqua [ORCID] http://orcid.org/0000-0001-5703-8382

**Decision letter and Author response**
Decision letter https://doi.org/10.7554/eLife.80009.sa1
Author response https://doi.org/10.7554/eLife.80009.sa2

## Additional files

**Supplementary files**
• Supplementary file 1. The file contains supplementary tables arranged in separate sheets, from A to G. (A) Complete Information for the teff samples used in this study. For each accession, the table reports the name of the accession at the germplasm repository (Name), the ID used in this study, the species, the type of materials (Status), the pedigree if known, the source of the accession, research center (Center) and release year if available (EBI, Ethiopian Biodiversity Institute; DZARC, Debre Zeit Agricultural Research Center; AARC, Adet Agricultural Research Center; HARC, Holleta Agricultural Research Center; BARC, Bako Agricultural Research Center; MARC, Melkassa Agricultural Research Center; SARC, Sirinka Agricultural Research Center). For accessions deriving from landraces, the table also reports the corresponding agroecological zone and the region name at three levels (Region_1, Region_2, and Region_3), the longitude, and the latitude. The table then reports the genetic cluster assigned by the analyses, the altitude, and the historical bioclim variables. Finally, BLUP values for phenotypes are farmer traits are reported. When specific by location, the trait code is attached to either Akaki or Adet. When specific by gender, M (men) or W (women) is attached. DH, days to heading; DM, days to maturity; PH, plant height; PL, panicle length; PBPM, number of primary branches per main shoot panicle; TT, total tillers; CLF, first culm length; CDF, first culm diameter; PW, panicle weight; PY, panicle yield; GY, grain yield; BY, biomass yield; HI, harvest index; GFP, grain filling period; GFR, grain filling rate; BPR, biomass production rate; OA, overall appreciation; PA, panicle appreciation. (B) Haplotype blocks for SNPs used in this study. For each SNP, the table reports the SNP name, the chromosome, and the position. If any, the table reports the corresponding haplotype block with start and stop position and number of SNPs in the block. (C) Variance components from BLUP model calculation on agronomic traits and farmer traits. The table reports solution, standard error, $z$ ratio, and percent variance explained for each component of the model for each trait. Trait codes as in Supplemenary File 1A. Factor codes as follows: ID, genotype; REP, replication within the field; LOCATION, field location; F_TYPE, gender; interactions between factors are indicated as in F_TYPE:LOCATION, for example gender by location. (D) Main effects of and interactions between location, genetic cluster, and, in the case of PVS traits, gender, regarding teff traits. Phentoype codes as in (A). Values in the table are p values for a two- (metric traits) and three-way ANOVA (PVS traits). Significant effects are highlighted in bold. (E) Genome-wide association results. For each SNP-trait combination, the table reports the trait name, the SNP name, its chromosome, and position. The reference (REF) and alternative (ALT) allele are reported for each SNP. For each association, the table reports the effect, the standard error (SE), the corresponding p value (pvalue), and multiple-test correction with a $q$ value (qvalue). The number of PC covariates used in the GWAS scan is reported in column n_PC. DH, days to heading; DM, days to maturity; PH, plant height; PL, panicle length; PBPM, number of primary branches per main shoot panicle; TT, total tillers; CLF, first culm length; CDF, first culm diameter; PW, panicle weight; PY, panicle yield; GY, grain yield; BY, biomass yield; HI, harvest index; GFP, grain filling period; GFR, grain filling rate; BPR, biomass production rate; OA, overall appreciation; PA, panicle appreciation. Bio2: mean diurnal temperature range; Bio3: Isothermality; Bio4: Temp. Seasonality; Bio9: Mean Temp. of Driest Quarter; Bio13: Precipitation of Wettest Month; Bio14: Precipitation of Driest Month; Bio15: Precipitation Seasonality; Bio18: Precipitation of Warmest Quarter; Bio19: Precipitation of Coldest Quarter. PC1_bio: first bioclimatic principal component; PC2_bio: second bioclimatic principal component; PC3_bio: third bioclimatic principal component. (F) Candidate genes mining for significant associations. The table reports genes in LD blocks with at least one significant association. Each row reports the *Eragrostis tef* gene ID (Et_gene), with chromosome, start position, end position, and DNA strand (positive + or negative −). When present, the table reports the closest homolog gene in *Arabidopsis* (id_At) with the percentual identity in the alignment (perc_identity_At), the *E* value reported by the BLASTP, the percentual query coverage in the target gene (perc_query_coverage_per_subject_At). The same information is reported for *Zea mays* (Zm) hits. The table reports the name of the LD block and the number of quantitative trait loci (QTN) in that block. (G) SNPs with adaptation potential. For each marker, the table reports the

chromosome, position, $F_{st}$ values, and gradient forest model fit (GF $r^2$).

• MDAR checklist

### Data availability

Teff accessions are available upon request from the Ethiopian Biodiversity Institute (EBI, http://www.ebi.gov.et/). Raw DNA sequencing reads are available on the Short Read Archive (https://www.ncbi.nlm.nih.gov/sra/) under BioProject accession number PRJNA758057. All scripts used for data analysis are available on GitHub at https://github.com/mdellh2o/TeffDiversityPanel, (copy archived at swh:1:rev:4935b0ef76f4c1631d323d39777ec375912e2f82).

The following dataset was generated:

| Author(s) | Year | Dataset title | Dataset URL | Database and Identifier |
|---|---|---|---|---|
| Dell'acqua M | 2021 | Genetic characterization of Ethiopian Teff diversity panel | https://www.ncbi.nlm.nih.gov/bioproject/PRJNA758057 | NCBI BioProject, PRJNA758057 |

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
