## [Editor Report]

Teff (*Eragrostis tef*), a small-market domesticate native and commonly grown in Ethiopia and the Horn of Africa, is comprehensively characterized for genetic, ecological and phenotypic variation in this ambitious and interdisciplinary publication. Integration of small holder farmers in phenotyping the collection, with an emphasis on gender considerations, elevates the characterization of Ethiopian teff. This paper provides a solid foundation to accelerate teff breeding for a changing climate, and provides an excellent model for the characterization of novel and underused crops.

---

## [Decision Letter]

**Decision letter after peer review:**

[Editors’ note: the authors submitted for reconsideration following the decision after peer review. What follows is the decision letter after the first round of review.]

Thank you for submitting the paper "Data-driven, participatory characterization of farmer varieties discloses teff breeding potential under current and future climates" for consideration by *eLife*. Your article has been reviewed by 3 peer reviewers, one of whom is a member of our Board of Reviewing Editors, and the evaluation has been overseen by a Senior Editor. The following individuals involved in review of your submission have agreed to reveal their identity: Bela Teeken (Reviewer #2); Laura Morales (Reviewer #3).

Comments to the Authors:

We are sorry to say that, after consultation with the reviewers, we have decided that this work will not be considered further for publication by *eLife*.

There was a lot of enthusiasm for the work in principle, but the common judgment was that revision would be considerably more extensive than commonly expected for *eLife*. Nevertheless, while we cannot formally invite revision, we remain very interested in the work. If you resubmit a suitably revised version, it would be treated as a new submission, but we would aim to recruit the same editors and reviewers to critique the work.

All of the reviewers were enthusiastic about the comprehensive scope of the manuscript but they raised concerns regarding a number of the analyses. The reviewers make specific recommendations regarding these concerns that should be addressed in a future submission. Critically, two of the reviewers highlight methodological issues with the choice of 6 clusters that were used for the diversity analyses that underlie a number of the subsequent analyses.

*Reviewer #1:*

Overall summary

"Data-driven, participatory characterization of farmer varieties discloses teff breeding potential under current and future climates" by Woldeyohannes and colleagues is a fascinating, interdisciplinary approach for understanding genetic, ecological and phenotypic variation in teff (Eragrostis tef), a small-market domesticate native and commonly grown in Ethiopia and the Horn of Africa. They assess genetic variation in traditional (landrace) and improved breeding varieties from a germplasm collection representing the diverse ecological zones of Ethiopia, attempting to link genetic with environmental variation. The collection was evaluated in two diverse locations by the authors for breeder phenotypes but also by smallholder teff farmers. Heritability for farmer traits is high and also highly correlated with breeder traits. Interestingly, men and women, while they mostly agree on the best varieties, have slightly different preferences. The authors conduct a GWAS but present very few results, mostly based on a few randomly chosen QTN. Finally, the authors present an interesting analysis using a gradient forest predictive model to understand the impact of environmental gradients of genomic variation, which will help identify vulnerable areas under climate change. This is an important topic that is outside of my field.

While the data collection, methodology and aims of the paper are very compelling, it suffers from some important interpretation and analytical issues in the characterization of genetic diversity. Specifically, clustering analysis used throughout the paper was not appropriate based on the data presented. The authors present some tantalizing results associating genetic diversity of environmental diversity, but these analyses could be clarified, tightened and streamlined. The authors attempt characterize genetic diversity in terms of social interactions but I did not find the proxy used for social interaction convincing. The phenotypic analysis, especially with respect to farmer selections was very clear and compelling but it would have been nice to see more follow-up on the GWAS analyses associated with these traits.

Introduction

The authors do a good job of introducing the market importance of teff and its role in small-holder agriculture but I would have liked more information on teff biology. Line 89 states that it's a tetraploid, but is it auto or allo? Is the ancestor also tetraploid? What was it domesticated from? Naturally outcrossing or inbreeding? What were historical effective population sizes (in the landraces and the breeding material)?

Results

Teff farmer varieties harness broad genetic diversity

This section showcases the genetic relationships between accessions and is associated with figure 1. The genetic material includes landraces, breeding lines and outgroup wild relatives on RADseq markers, which is only clear from looking at the associated methods and a summary here would be welcome. While the Admixture and PCA analyses are performed correctly, there are serious issues with the interpretation of outputs.

Admixture results (K = 2-10) are presented as well as a PCA plot colored based on the DAPC clustering for the same PCA. The DAPC clustering results are then shown in geographic space, according to altitude and to agroecological zone.

Buried in supplementary material 4 are the metrics for evaluating model fit for the admixture as well as the DAPC clustering. The lowest cross-validated error rate (best predictive accuracy in leave-one-out) is for K = 20, but the authors state the K = 6 is the "best" K. Likewise, the lowest BIC (model fit) value for the DAPC clustering is 10, not 6 as the authors state.

The authors decision to promote six clusters is not supported and the main figure 1 and all subsequent uses will need to be updated to reflect the best fit values (DAPC 10; admixture 20). It would also be informative to contrast breeding material with landraces as this is one of the main aims of the study.

The distribution of teff genetic variation is associated with geographic and environmental factors.

This section, associated with Figure 2, seeks to contextualize teff genetic variation within environmental and social variation. Overall, the aim of contextualizing genetic variation ecologically and socially is a good one and the ecological associations are interesting and appropriate but I don't find the social proxies convincing and some of the analyses – the neighbor joining plot with outgroups in S2, and perhaps the FST analysis in 2E – are out of place.

2A is a PCA of bioclimatic variation from interpolated weather data assigned to each landrace and there are interesting patterns with respect to DAPC cluster groups (which will need to up updated), which are highlighted in the text. 2-C and D show FST values, presumably based on populations assigned by regional political district as in part E but this is not explained, against geographic and environmental (the average of orthogonal bioclimatic variables and altitude) difference. The regional districts are supposed to be a proxy for social connections and markets (lines 185-187) but there is no citation or presented evidence for why this is a reasonable proxy.

The main conclusion (lines 188-192), that both isolation by geographic distance and environmental distance contribute to genetic differentiation, is not surprising to since the populations used for FST were defined geographically and the geographical and environmental distance are likely fairly highly correlated. 2E shows the F¬ST matrix, which shows that samples from Tigray are the most genetically isolated, while those from Amhara and Oromia are relatively more admixed. F2-S1 are boxplots of the bioclimatic values clustered by the DAPC groups, which, with updated DAPC clustering, are an interesting look at genetic and spatial differentiation, especially the bottom row with the values for the bioclimatic PCs.

Participatory evaluation of the teff diversity prioritizes genetic materials for breeding

This section, associated with figure 3, explores the participatory evaluation results, comparing scores between men and women and between the farmer and breeder evaluations. There was generally very high agreement between breeders and farmers and some intriguing insight into gender-based differences in evaluation criteria. The phenotypic models were appropriate and data presented and textual discussion was overall clear and well-supported.

Participatory, climatic, and agronomic diversity identify candidate loci for teff breeding

The approaches used for GWAS analysis are appropriate based on that reported in the methods but there are no supporting figures. At the least, I would expect Q-Q plots to assess population structure correction and Manhattan plots in the supplemental. For the specific genes that the authors focus on it would be good to also see the local Manhattan and LD information. However, the litanies of linked gene models could also probably be dropped or minimized in the text (it may still be an important resource for future studies to have in the public record). I think the authors could have done more with the GWAS results.

Teff cultivation is vulnerable to climate change

This section, associated with figure 4, uses a gradient forest predictive model to understand the impact of environmental gradients of genomic variation, which will help identify vulnerable areas under climate change. These are not analyses I am familiar with and I do not feel qualified to review this section in detail.

Conclusion

The conclusions are pertinent and help motivate future directions.

Methods

The methods are generally well described. As the methods are situated after the results, it would be useful to reiterate some of the key points from the methods under results to help situate the reader.

Specific comments and suggestions

It would be good to give the whole manuscript a copy edit. There were a number of small grammatical and word choice issues I noted, a few of which are below, along with additional suggestions.

127 what is the difference between moist and humid? Citation?

132 what kind of libraries? There should be a short statement of library choice and associated biases.

163-165 This statement is probably true if you're deeply familiar with the germplasm and the environments, but it's not clear to someone with no prior knowledge based on the figure. It's actually better supported by figure 2A.

174-175 Figure S1-5 does not support this statement. Beyond the fact that the K was chosen at random, as were the DAPC clusters, admixture is not quite the right evidence. A better line of support for this statement would be something along the lines of "Limited population stratification, as evidenced by less than 15% of the total genetic variance explained by the first three principal components (Figure 1-B-C)…".

200-204 You can refer back to other figures to connect dots – it would make more sense for the pop gen to be presented together.

207 What environments are these trial locations in? More description.

218-220 Define shorthand phenotypes here (the ones used in figure 3)

Figure 1: It is not completely clear from the legend in Figure 1, but I believe these figures include the landraces and breeding material (except panel D – this must be clarified). I would suggest that rather than the current panels it might be more informative to just see admixture (K = 20) ordered by breeding vs landrace, altitude or agroecological zone rather than based on unsupervised clustering (admixture). Likewise PCA plots colored based on independent values (altitude, improvement status, etc) may be more informative than using the DAPC cluster assignments (K = 10), although I like Figure S1-6 and the boxplots in 1E are probably more informative than just coloring based on altitude would be. There is nothing about improvement status in this figure – it would also be interesting to highlight the genetic relationships between the breeding and landrace material (S3 suggests that much of the breeding material clusters)

Figure 2: I would rethink the social proxy (market analyses? Ethnicity?) and clarify the analyses performed for the bioclimatic associations. I think S1 (the last row) would be a good analysis to include in the main figure. I would put S2 with Figure 1, maybe I'm missing something but I'm not sure why it's there. I also don't understand why the correlations between the climate variables and the PCs are presented in B – this is clearly visible from the eigenvectors in A – but if it is informative I would suggest more discussion.

Figure 3: Very interesting. It might be interesting to see the correlation plot in A separated by men and women.

What happened to the GWAS analysis?

Methods:

Plant Materials: seems reasonable and sufficiently described.

Sequencing and Variant Calling: I'm not very familiar with RADseq calling pipelines but it seems reasonable. This is a well-established approach though and if the authors are following a previously published processing pipeline it would be good to cite.

Spatial and bioclimatic characterization: It's not clear to me when the different datasets are used, and why you would use one over the other.

470 Why use the highest resolution for something like a landrace that has a km rather than meter resolution?

More information on the grow-out locations would be welcome.

*Reviewer #2:*

In the context of teff production and consumption in Ethiopia the authors advocate breeding initiatives to strengthen this crop that is important for the livelihoods of the many small holders in Ethiopia by highlighting the limited yields, the limited breeding efforts made in this crop and the anticipated impactful climate change. By using genetic, climatic, geographic and participatory variety selection, gene pools are identified that would best inform parents to use for future breeding initiatives. This goes even further and shows the possibility to identify crucial genetic markers needed for breeding focus.

With convincing strength, the authors employ genetic analysis of landraces collected all over the country and link this to the different agro-ecological regions and niches as well as the different ethnic traditions and to the climatic data including a forecast of climatic factors based on an extrapolation of historic climate data. Although overwhelming at first, the large amount of figures well illustrate the argument and well satisfy if one wants to get some deeper understanding (supplementary figures). A great innovative strength is the proposition of a standard toolbox to combine the different data to provide product profiles for teff breeding.

The authors rightly state that the suitability of varieties does not only depend on ecological, climatic and product data but also on socio-cultural variables and they take this into account by stating that the regions coincide with the different ethnic groups in the country. These ethnic groups could however be named and their differences with regard to teff related practices and preferences highlighted and related to the two different groups of farmers chosen for the participatory variety selection. The selection strategy of the farmers could also be provided.

A little more discussion could be provided on the representativeness of the two PVS and phenotyping sites and the farmers evaluating those. In the conclusion the authors cite the TRICOT approach as a way forward to decentralize PVS and get better systematic and representative testing of the genetic resources. The limitation of the only two sites in which the genetic collections were grown and the possible G x E influence on the phenotyping of the varieties could also be highlighted. However, the fact that with the two locations such great results were obtained in linking all the datasets shows the potential of the approach taken.

321 landraces from 3850 Provide were selected for the study. It will be important to provide the selection strategy used to get to the 321 varieties. This selection strategy could also be highlighted outside of the methodology. The selection strategy might also be related to the time of collection of the varieties. Are these 321 chosen in a way to simply represent the largest genetic diversity in the collection or is the time of collection of the samples also included: to represent the varieties that are currently still cultivated?

With regards to choosing men and women and comparing them separately there is not discussion on why one would expect differences. Only in the conclusion Weltzien et al. is cited in this respect but this needs more attention and reference to some more literature, especially in relation to the Ethiopian context. What is especially needed is an explanation of the different tasks (roles) of men and women in Ethiopia with regards to teff cultivation and possibly processing. This should be provided in general and in relation to the two study sites. This could also provide the background to a short discussion to interpret the similarities and the difference observed (appreciation being more related to biomass yield and panicle weight for women while for men it was plant height and grain filling period, could this be explained by women using plant parts for animal feed or any other activity in which they more feature?). I know that this is not the main focus of the paper but the fact that men and women's preferences were analyzed and discussed separately demands a discussion with regards to the results and the strategy used to select the men and women.

This work is not only relevant to under NUS crops but also for breeding in general and especially for public sector breeding. In this respect the authors could open up this approach to be more generally applied to other crops and could highlight the importance of their approach for customer and product profiling and especially for customer profiling with which especially public breeding is tasked to achieve social impact as cost effective as possible.

Figures

With regards to figures 1 and 2: it is great to have all these figures close to each other for good cross referencing, however in that case it will be very important to have extremely good resolution. A solution could be to arrange the figures in a way that the now smaller figures in Figure 1 become bigger and the larger figure (Figure 1 D) become a little smaller so that the PCs are better visible, this is minor work to arrange. Also, the legenda with the colored dots indicating the DAPC clusters could be made much bigger for easier determination of the colors. The same counts for figure 2.

Please verify the Y axe on figure 3E. Should this not be PA instead of PL? It was panicle appreciation that was evaluated with the farmers not panicle length?

*Reviewer #3:*

The authors have conducted an interdisciplinary characterization of a valuable teff diversity panel using well-supported methods, except for the selection of genetic clusters. Although the methodology itself is not novel, the interdisciplinary nature of the study sheds light on a wide range of Ethiopian teff germplasm characteristics, from environmental adaption to gender preference.

Strengths:

The incorporation of farmer's knowledge and gender preferences in the genotypic and phenotypic analysis of the germplasm was well done and nicely presented. This kind of information is absolutely necessary for the improvement and intensification of NUS, especially in smallholder farmer systems. Few germplasm characterization studies, both on NUS and staple crops, have reported such a broad range of characteristics.

Weaknesses:

Although many of the analyses rely on the differentiation of genetic clusters, the authors have inappropriately selected the optimal number of clusters (K). The authors conducted a cross-validated error analysis of various ADMIXTURE K values and a similar analysis of BIC statistics of various DAPC K values. Figure 1—figure supplement 4 clearly demonstrates that K=20 for ADMIXTURE and K=10 for DAPC, as these values of K have the lowest cross-validated error and BIC, respectively. However, the authors claim to have chosen K=6 because (a) there is a slight flattening of the ADMIXTURE error curve after K=6 and (b) K=6 has the lowest DAPC BIC statistic, neither of which are supported by the results.

Although the authors reported high broad-sense heritability (H2) for several agronomically important traits, variance components for the terms (genotype, environment, GxE, gender, error, etc.) used to estimate H2 were not shown. A table including H2, variance components, and the numbers of levels and observations for each term is necessary to assess the validity of these results. This information would also shed light on the relative influence of genetics, environment, gender, etc. on trait variation.

As stated above the methods used to select K=6 as the optimal number of clusters was statistically inappropriate. Although K=20 had the lowest cross-validated ADMIXTURE error, this value seems too high. I would suggest using K=10, as this value of K had the lowest DAPC BIC, showed a more clear flattening in the ADMIXTURE error curve, and is closer to your desired value of K=6. Did K=6 correspond to some prior knowledge on the diversity panel, for example from breeders' knowledge? If so, perhaps this could be stated or incorporated in some way to support your choice of K=6.

[Editors’ note: further revisions were suggested prior to acceptance, as described below.]

Thank you for resubmitting your work entitled "Data-driven, participatory characterization of farmer varieties discloses teff breeding potential under current and future climates" for further consideration by *eLife*. Your revised article has been evaluated by Detlef Weigel (Senior Editor) and a Reviewing Editor.

The manuscript has been improved but there are some remaining issues that need to be addressed, as outlined below:

Essential revisions:

1) Further explicate the relative contributions of genotype, environment and GxE for the traits evaluated, per the more detailed suggestions of reviewers 1 and 3. This may be supported by the Oryza research suggested by reviewer 2.

2) Reassess the discussion of the GWAS results in the text and supporting material, particularly DM and PC2_bio. Per comments by reviewer 1, the QQ plots show that these traits are highly impacted by residual population structure and thus subject to an excess of false positives.

3) Provide a bit more discussion of gendered roles in agricultural production, taking into account comments by reviewer 2.

*Reviewer #1:*

The authors in their responses have addressed most of my concerns and I think the paper reads much more clearly now. I only have two additional comments:

With respect to the GWAS results, these can now be evaluated with the inclusion of the QQ plots. All traits except bio14 are somewhat confounded with population structure (genome-wide deviation from expectation of no association, or deviation from the line in the QQ plot) and DM and PC2_bio are very correlated with structure and the QQ plot suggests that neither of these traits have truly associated loci in this study. The authors discuss results from especially DM a great deal and I think the discussion of these results should be reevaluated.

The authors also state in the conclusion that "Multi-location trials are needed to capture the range of genotype by environment (G x E) interactions that influence agronomic performance of teff, which we could only partially characterize" but they do have two locations and are able to model GxE (although, more locations would certainly lead to better estimates). For a genomic resource paper, it would be very beneficial to report the impact of GxE directly (and how the landraces and improved varieties may differ, as has been found in other crops).

*Reviewer #2:*

The authors have done a great job in revising the manuscript, which is much easier to read and figures and tables. The organisation has also improved. As far as my expertise allows the authors have also well approved the technical analysis part of the manuscript.

The sampling of the PVS participants has been described satisfactorily and the reason why to include gender has been articulated. I can imagine that the authors did not find any literature or resources to explain the difference observed between women and men. I highlighted that a reason could be that women are tasked with animal feeding and look at the plant's vegetative parts also as a resource of animal food while men would not focus on that so much because it might be out of their gendered roles. The authors could quickly consider this issue and see if there is any probability in this being the case or any other reason that could be mentioned to explain the differences observed between women and men. Any little clue or hypothetical phrase would give the gender dimension just a little more depth and clue for further investigation, rather than just saying that differences are to be expected.

The authors nicely made the link to general breeding and the development of customer and product profiles in (public) breeding that are more and more expected to deliver on social impact in line with the sustainable development goals

The issue I raised with regards to the representativeness of the PVS trial locations has also been resolved satisfactorily.

This is now a really strong paper and needs to be published so I will be able to share it as a good example of transdisciplinary data driven research.

Please correct on line 161-162: reference is made to two figures (as the word 'and' is written) but only figure 6 is mentioned.

Furthermore, I would like to point the authors to the following publications based on work on the under researched African rice (Oryza glaberrima) and how farmers long trajectories of selection resulted in selecting 'robust' varieties that are adapted to change and dynamics (social as well as climatic) rather than narrow local localities. Any reference to this body of work could be made, although the manuscript stands strong as it is without referring to this body of work.

http://journals.plos.org/plosone/article?id=10.1371/journal.pone.0034801

http://journals.plos.org/plosone/article?id=10.1371/journal.pone.0085953

http://journals.plos.org/plosone/article?id=10.1371/journal.pone.0007335

https://link.springer.com/article/10.1007/s10745-012-9528-x

*Reviewer #3:*The authors have done a nice job revising the manuscript using 10 genetic clusters, following the recommendations of the reviewers.I would suggest incorporating the information in Supplementary file 1C into Tables 1 and 2. For example, columns for variance explained by genotype, location, rep, error, etc. could be added to these tables.

I also recommend discussing the results in Supplementary file 1C (or Tables 1-2, if the authors choose to merge this information) further. The experimental design appears to have allowed for high estimation of heritability, which the authors have already stated in the first and latest versions of the manuscript. However, as a plant breeder, I would like to know more about the relative contributions of genotype, environment, GxE on the different traits measured in this germplasm. For example, it is interesting that (a) for BPR, location explains 75% of the variance, while genotype explains a 18% of the variance and there is no GxE effect, which contrasts to (b) CDF, which has large GxE variance (66%) and relatively smaller proportions of the variance are explained by genotype (7%) and location (19%). There are likely some trends/comparisons among traits that can be discussed, such as which traits tend to have high GxE vs traits that have high location effects vs traits with high genetic variance.

[Editors’ note: further revisions were suggested prior to acceptance, as described below.]

Thank you for resubmitting your work entitled "Data-driven, participatory characterization of farmer varieties discloses teff breeding potential under current and future climates" for further consideration by *eLife*. Your revised article has been evaluated by Detlef Weigel (Senior Editor) and a Reviewing Editor.

The manuscript has been improved but there are some remaining issues that need to be addressed, as outlined below:

1) Both reviewers 1 and 3 are still concerned with the interpretation of the GWAS. Please see especially reviewer 1's lengthy explanation to help clarify interpretation. Please take these concerns into account when interpreting GWAS results and reporting significant loci.

2) All of the reviewers have made small, specific comments that should be addressed.

3) A final copy-edit after changes have been made.

Reviewer #1:

The paper reads much better and I am satisfied with the incorporation of GxE analysis. I still have concerns with the interpretation of the GWAS results and would recommend a final copy edit to fix the grammatical errors introduced in the editing process.

Statistically significant p-values are detected as deviations from the expectation of no-association based on statistical linkage between the causal association and the tested SNP (there are likely no causal loci tested in this limited marker set). The foundational assumption in GWAS is that these associations will be in local LD based on limited recombination in the region surrounding the causal polymorphism, allowing the researcher to zoom in on the loci underlying trait variation. This is visualized on an QQ plot as most dots on the diagonal of no-association with a small deviation associated with only the most significant SNPs (bio14 is a good example of this). On a Manhattan plot, these will for a localized peak around causal loci, characterized by the extent of the haplotype block associated with the causal polymorphism and the sampled marker density.

However, linkage can also exist between causal polymorphism and loci across the genome, even on other chromosomes, because selection is not random or evenly applied across a species or population. When populations differ due to drift processes, and selection is nested within randomly varying genetic structure, GWAS tests for the trait under selection will also find associations with the variants associated with differentiation between populations. For example, one population lives in a cold, highland environment and one lives in a tropical, lowland context. The highland population has a shorter growing season and consequently flowers earlier to ensure that the grain can mature in time before frost. The populations were already a bit different due to distance and chance, but now that the flowering window is non-overlapping for these populations it ensures that gene flow is basically stopped and that the populations will move further apart. If one were to do a GWAS for days to flowering or maturity (or for temperature or growing season variation) across these two populations the resulting QQ and Manhattan plot would look like the ones in the analysis for PC_bio2 or for DM; an early a persistent deviation from the expectation of no association in the QQ that localizes all over the genome in a Manhattan plot. This is not to say that there are not real associations (the peak on 6A is likely associated with a real causal locus for DM), but they are confounded with underlying structure and "statistically significant" associations are no longer a good way to identify truly associated loci. A better way to think of GWAS with these qualities is that the top SNPs are enriched for linked causal associations but any given association is suspect.

With this in mind, I would suggest a bit more discussion of confounded structure. Perhaps a way to frame it is in the context of local adaptation, tying in the GF models and variance analysis of GxE for the various traits.

331-336: The accepted associations for DM and PC2_bio are still very lax and based on the QQ plots heavily confounded with underlying population structure. This underlying structure is almost certainly driving association with the gradient forest model and I think that the conclusion that this association "support the importance of phenology in teff adaptation and geographic distribution" is reasonable, but not because of genetics underlying days to maturity per se. If the authors were to include a statement to this effect the overall interpretation is reasonable.

341-343: Again, when two traits are both heavily confounded with underlying structure it is not surprising that they would colocalize (via the third variable of structure), and I would be especially sceptical of colocalized regions between these traits.

349-355: Agree that this peak is likely real because of the very strong local LD. Not convinced of anything else

Reviewer #3:

The authors have done a nice job including more information about GxE, genetic variance, etc.

I am still not convinced of the GWA results for DM and PC2_bio, which show a very large deviation from the expected p-value distribution. The abundance of false positives cannot merely be dismissed with the authors' statement that "some of the associated traits…showed some statistical inflation on QQ plots likely contributed by residual population structure". A large proportion of all SNPs tested were deemed significant (assuming that the significance cut-off was approximately -log10p = 3.5) for DM and PC2_bio. I would suggest that the authors use a secondary threshold based on a visual assessment of the QQ plots. For DM, I would suspect that the ~5 SNPs with -log10p > 4.8 are truly significant (QTL on chromosome 6A). Similarly for PC2_bio, the 2 SNPs at the tail of the distribution are likely significant.

With respect to the gender aspects of this manuscript, I would recommend including references from similar work done by the NextGen Cassava project. Here are two examples:

https://doi.org/10.1007/s12231-018-9421-7

https://doi.org/10.1002/csc2.20152

---

## [Author Response]

[Editors’ note: the authors resubmitted a revised version of the paper for consideration. What follows is the authors’ response to the first round of review.]

Reviewer #1:IntroductionThe authors do a good job of introducing the market importance of teff and its role in small-holder agriculture but I would have liked more information on teff biology. Line 89 states that it's a tetraploid, but is it auto or allo? Is the ancestor also tetraploid? What was it domesticated from? Naturally outcrossing or inbreeding? What were historical effective population sizes (in the landraces and the breeding material)?

We added more details about teff biology in the Introduction section, adding two new references (see below). Being an autogamous species, the effective population size *Ne* can be computed as half the number of individuals.

“Teff (Eragrostis tef) is an annual, self-pollinating, and allotetraploid grass of the Chloridoideae subfamily (Ketema, 1997). It is widely grown in Ethiopia as a staple crop, where it is valued for its nutritional and health benefits, resilience in marginal and semi-arid environments and cultural importance (D’Andrea, 2008; Ketema, 1997). Teff was likely domesticated in the northern Ethiopian Highlands from the allotetraploid Eragrostis pilosa, but the timing of its initial cultivation and the identity of its diploid ancestors remain poorly understood (D’Andrea, 2008; Ingram and Doyle, 2003; VanBuren et al., 2020).”

ResultsTeff farmer varieties harness broad genetic diversityThis section showcases the genetic relationships between accessions and is associated with figure 1. The genetic material includes landraces, breeding lines and outgroup wild relatives on RADseq markers, which is only clear from looking at the associated methods and a summary here would be welcome. While the Admixture and PCA analyses are performed correctly, there are serious issues with the interpretation of outputs.

We revised and clarified this section so to aid readability:

“The EtDP comprises 321 farmer varieties spanning the entire geographical and agroecological range of teff cultivation in Ethiopia, from the sub-moist lowlands of Tigray in the North to the moist lowlands of Oromia in the South, and from the sub-humid lowlands of Benishangul and Gumuz in the West to the sub-humid mid-highlands of Oromia in the East (Figure 1 —figure supplement 1, Supplementary file 1A) (MoA, 2000). The EtDP also includes 45 improved varieties released since the first breeding efforts on teff and up to the moment of the EtDP creation. A selection of seven Eragrostis spp., putative wild relatives of teff, was included as outgroup. The genomic diversity of the EtDP was assessed starting from 12,153 high-quality, genome wide single nucleotide polymorphisms (SNPs) derived from double digest restriction-site associated sequencing (ddRAD-seq) of individual accessions, followed by filtering for variant call quality and linkage disequilibrium (LD) pruning.”

We revised Figure 1, Figure 3 and all Figures were relevant so to highlight the separation between farmer varieties and breeding materials with different symbols.

Admixture results (K = 2-10) are presented as well as a PCA plot colored based on the DAPC clustering for the same PCA. The DAPC clustering results are then shown in geographic space, according to altitude and to agroecological zone.Buried in supplementary material 4 are the metrics for evaluating model fit for the admixture as well as the DAPC clustering. The lowest cross-validated error rate (best predictive accuracy in leave-one-out) is for K = 20, but the authors state the K = 6 is the "best" K. Likewise, the lowest BIC (model fit) value for the DAPC clustering is 10, not 6 as the authors state.The authors decision to promote six clusters is not supported and the main figure 1 and all subsequent uses will need to be updated to reflect the best fit values (DAPC 10; admixture 20).

The reviewer is right in that the best interpretation for the DAPC clustering is 10. The reason why we initially settled on DAPC 6 is that this was the most parsimonious interpretation that could be provided to the clustering procedure. However, we take the point, and we have changed our clustering interpretation throughout the whole manuscript, discussing the existence 10 clusters. This resulted in update figures and updated text, as well as in a change in interpretation of the results. All changes, that are too many to be reported in this file, are highlighted in the attached revised manuscript text

It would also be informative to contrast breeding material with landraces as this is one of the main aims of the study.

Indeed. To address this point, we added a more thorough discussion of the clustering outcome for improved varieties and landraces and changed the figures to better depict the type of genetic materials. Changes are widespread in text and include:

“Breeding lines showed a narrower genetic base compared to the diversity maintained by Ethiopian farmers; these materials could not represent the broad diversity available within the EtDP (Figure 1 —figure supplement 4), and predominantly belonged to clusters 2, 8 and 10 (Figure 3D-E).”

The distribution of teff genetic variation is associated with geographic and environmental factors.This section, associated with Figure 2, seeks to contextualize teff genetic variation within environmental and social variation. Overall, the aim of contextualizing genetic variation ecologically and socially is a good one and the ecological associations are interesting and appropriate but I don't find the social proxies convincing and some of the analyses – the neighbor joining plot with outgroups in S2, and perhaps the FST analysis in 2E – are out of place.

We agree that a clear-cut social interpretation of teff diversity is not supported by the available, and we removed the outgroups in the NJ plot in S2. We maintained however the Fst plot, as it may be useful for audience knowledgeable of the Ethiopian geography (*e.g.* teff breeders) to support prioritization of genetic materials by local breeders. The point that the reviewer raises is critical: genetic variation in smallholder farming systems must be contextualized in ecologic and social dimensions. Our approach is speculative, but supported by previous literature now cited and extensively discussed:

“Extant landraces diversity is not only contributed by climate, but also by seed circulation. Studies have shown that cultural and social dynamics, such as belonging to similar or different ethnolinguistic groups, are key factors in shaping seed exchange networks (Labeyrie et al., 2016; Samberg et al., 2013). In Ethiopia, regional districts are markers of cultural and historical diversity, and smallholder farmers have limited capacity for long-distance travel. Thus, we hypothesized that regional distinctions could impact seed exchanges and spatial patterns of teff genetic diversity. The district of sampling of teff landraces in the EtDP was therefore used to aggregate accessions and calculate genetic distances as a measure of fixation index (Fst). Although our experimental design does not allow to fully untangle the effect of geography, environment, and social factors, we found a pattern of genetic variation distribution that can be associated to an isolation by distance and adaptation process, where Fst values are significantly associated with geographic distance (Mantel r = 0.31; p = 9e-04) and environmental distance (Mantel r = 0.352; p = 0.0137) (Figure 2C-D).”

We rephrased the Conclusion paragraph to better discuss perspectives in teff improvement, stressing the need for a socioeconomic characterization of seed systems

“An extensive socioeconomic characterization of teff cropping, including extensive farmers interviews (Labeyrie et al., 2016), may further unveil the dynamics of teff seed exchange and associated flow of information and knowledge related to local cropping (Occelli et al., 2021). Public sector breeding beyond NUS may further enhance the combination of data-driven research with participatory approaches to improve customer and product profiling to achieve social impact as cost effective as possible.”

2A is a PCA of bioclimatic variation from interpolated weather data assigned to each landrace and there are interesting patterns with respect to DAPC cluster groups (which will need to up updated), which are highlighted in the text. 2-C and D show FST values, presumably based on populations assigned by regional political district as in part E but this is not explained, against geographic and environmental (the average of orthogonal bioclimatic variables and altitude) difference.

Figure 2A has been updated with the new cluster interpretation. It is now clarified in text and legend how the Fst was computed for 2C and 2D, that is aggregating samples by local political district. The Figure legend was updated as follows:

“Figure 2. Teff diversity on the landscape. (A) Principal Component Analysis of bioclimatic diversity in the EtDP. Dots represent teff farmer varietiesf belonging to genetic clusters, colored according to legend. Vectors represent the scale, verse, and direction of bioclimatic drivers of teff differentiation. (B) Linear regression of Fst values in relation to geographic distance of accessions in the EtDP. Accessions were grouped by local district of sampling. (C) Linear regression of Fst values in relation to environmental distance of accessions, also grouped by district of sampling. (E) Pairwise Fst values between teff accessions grouped by local districts of sampling, as in (C) and (E). Local districts, i.e. sub-regional groups, are ordered by administrative regions according to legend. This figure has two figure supplements.”

The regional districts are supposed to be a proxy for social connections and markets (lines 185-187) but there is no citation or presented evidence for why this is a reasonable proxy.

This is a very good point and worth expanding in the text. We now provide a further explanation of the concept based on Labeyrie et al. (2016), who report how the belonging of ethnolinguistic groups impact seed exchange. In Ethiopia, studies shown that these same patterns involve different crops (Samberg et al. 2013) and we believe it is reasonable to assume that local culture, combined with limited capacity for long-distance travel, may affect local agrobiodiversity. This is expanded in text, as indicated above.

The main conclusion (lines 188-192), that both isolation by geographic distance and environmental distance contribute to genetic differentiation, is not surprising to since the populations used for FST were defined geographically and the geographical and environmental distance are likely fairly highly correlated.

It is true that the Fst was computed on populations that were geographically defined, but the patterns of gene flow are not always obvious. As discussed in other studies, there might be different gene flow scenarios, including IBD, IBE, counter-gradient gene flow, unrestricted gene flow, and limited gene flow (*e.g.* in Sexton, J. P., Hangartner, S. B., & Hoffmann, A. A. (2014) http://www.jstor.org/stable/24032844). We expanded this discussion in text, highlighting the reviewer criticism:

“Although our experimental design does not allow to fully untangle the effect of geography, environment, and social factors, we found a pattern of genetic variation distribution that can be associated to an isolation by distance and adaptation process, where Fst values are significantly associated with geographic distance (Mantel r = 0.31; p = 9e-04) and environmental distance (Mantel r = 0.352; p = 0.0137) (Figure 2C-D). Accessions from East Tigray (Misraquawi) showed the highest separation from the collection, followed by East Oromia (Misraq Harerge) and West Amhara (Agew Awi) (Figure 2E).”

2E shows the F¬ST matrix, which shows that samples from Tigray are the most genetically isolated, while those from Amhara and Oromia are relatively more admixed. F2-S1 are boxplots of the bioclimatic values clustered by the DAPC groups, which, with updated DAPC clustering, are an interesting look at genetic and spatial differentiation, especially the bottom row with the values for the bioclimatic PCs.

All figures, including the ones mentioned here, were updated considering 10 DAPC clusters

Participatory evaluation of the teff diversity prioritizes genetic materials for breedingThis section, associated with figure 3, explores the participatory evaluation results, comparing scores between men and women and between the farmer and breeder evaluations. There was generally very high agreement between breeders and farmers and some intriguing insight into gender-based differences in evaluation criteria. The phenotypic models were appropriate and data presented and textual discussion was overall clear and well-supported.

In this section, we updated figures and text to respond to reviewers’ remarks, specifically in relation to cluster number and relevance of improved varieties VS farmer varieties.

Participatory, climatic, and agronomic diversity identify candidate loci for teff breedingThe approaches used for GWAS analysis are appropriate based on that reported in the methods but there are no supporting figures. At the least, I would expect Q-Q plots to assess population structure correction and Manhattan plots in the supplemental. For the specific genes that the authors focus on it would be good to also see the local Manhattan and LD information. However, the litanies of linked gene models could also probably be dropped or minimized in the text (it may still be an important resource for future studies to have in the public record). I think the authors could have done more with the GWAS results.

Indeed, we performed a full GWAS analysis, with all outputs including Q-Q plots and Manhattan plots. In the first version of the manuscript, we left it aside as not to overload the items associated to the manuscript, however we accept the reviewer criticism and therefore we added new items and text describing the GWAS output in detail. We added a figure in the main text reporting a GWAS scan across PVS traits and metric traits, showing the co-mapping of a QTN for OA, yield and flowering time (now Figure 5). Associated to Figure 5, we included two supplementary figures, one depicting a GWAS scan focusing on environmental data, and one reporting Q-Q plots for reported associations.

For what concerns the description of QTNs in the main text: while the QTN that we focus on are a subset of all the QTNs detected, the selection is not made at random. Rather, we selected those QTNs that are more relevant for teff breeding in the light of previous studies on similar crops. We believe that this description may be useful for further studies exploring this and other teff gene pools, and thus we would prefer to keep it in the text. However, we revised the text to shorten and streamline the description of these gene models as suggested by the reviewer.

Teff cultivation is vulnerable to climate changeThis section, associated with figure 4, uses a gradient forest predictive model to understand the impact of environmental gradients of genomic variation, which will help identify vulnerable areas under climate change. These are not analyses I am familiar with and I do not feel qualified to review this section in detail.ConclusionThe conclusions are pertinent and help motivate future directions.MethodsThe methods are generally well described. As the methods are situated after the results, it would be useful to reiterate some of the key points from the methods under results to help situate the reader.

The text was revised to aid readability, reiterating relevant points in the Results section. The modifications are widespread in text and cannot be entirely reported here.

Specific comments and suggestionsIt would be good to give the whole manuscript a copy edit. There were a number of small grammatical and word choice issues I noted, a few of which are below, along with additional suggestions.

The manuscript was copy edited before resubmission.

127 what is the difference between moist and humid? Citation?

“Moist” has higher rainfall than “humid” and is found at higher altitudes. These are standard definition of agroecological zonation of Ethiopia, and all info is found in the cited document (*Agro-Ecological zones of Ethiopia*, 1998)

132 what kind of libraries? There should be a short statement of library choice and associated biases.

Fixed.

163-165 This statement is probably true if you're deeply familiar with the germplasm and the environments, but it's not clear to someone with no prior knowledge based on the figure. It's actually better supported by figure 2A.

The description for germplasm and environments have been improved

174-175 Figure S1-5 does not support this statement. Beyond the fact that the K was chosen at random, as were the DAPC clusters, admixture is not quite the right evidence. A better line of support for this statement would be something along the lines of "Limited population stratification, as evidenced by less than 15% of the total genetic variance explained by the first three principal components (Figure 1-B-C)…".

Fixed.

200-204 You can refer back to other figures to connect dots – it would make more sense for the pop gen to be presented together.

Fixed increasing cross references to figures.

207 What environments are these trial locations in? More description.

fixed adding a full description of the experimental sites in the Materials and methods section:

“The EtDP was phenotyped in common garden experiments in two high potential teff growing locations in Ethiopia, Adet (Amhara, 11° 16′32″ N, 37° 29′30″ E) and Akaki (Oromia, 8°50'07.6"N, 38°49'58.3"E), under rainfed conditions during the main cropping season of 2018 (July-November). Adet is the main experimental site of the Amhara Regional Agricultural Research Institute, at an altitude of 2,240 meter above sea level. Soil type is vertisol, climate is moist cool, and average annual rainfall is 1,250 mm. Akaki is a sub site of the Debre Zeit Agricultural Research Center, with an altitude of 2,200 meter above sea level. Also in this case, soil type is vertisol, climate is moist cool and average annual rainfall is 1,055 mm.”

218-220 Define shorthand phenotypes here (the ones used in figure 3)

Fixed.

Figure 1: It is not completely clear from the legend in Figure 1, but I believe these figures include the landraces and breeding material (except panel D – this must be clarified). I would suggest that rather than the current panels it might be more informative to just see admixture (K = 20) ordered by breeding vs landrace, altitude or agroecological zone rather than based on unsupervised clustering (admixture). Likewise PCA plots colored based on independent values (altitude, improvement status, etc) may be more informative than using the DAPC cluster assignments (K = 10), although I like Figure S1-6 and the boxplots in 1E are probably more informative than just coloring based on altitude would be. There is nothing about improvement status in this figure – it would also be interesting to highlight the genetic relationships between the breeding and landrace material (S3 suggests that much of the breeding material clusters)

We revised figure 1 according to these comments. The updated figure features the following: (A) ADMIXTURE results for the pruned SNPs dataset at values of K = 20 and now grouped as from the result of DAPC (10 DAPC clusters); In this figure the order of samples refers to results of complete linkage agglomerative clustering, based on pairwise identity-by-state (IBS) distance. (B) and (C) the PCA is now colored according to the DAPC cluster assignments (K = 10), varieties and landraces are represented with different shapes and the relationships between breeding materials and landraces can easily be observed. (D), (E) and (F) were amended and represented with 10 genetic clusters.

Figure 2: I would rethink the social proxy (market analyses? Ethnicity?) and clarify the analyses performed for the bioclimatic associations. I think S1 (the last row) would be a good analysis to include in the main figure. I would put S2 with Figure 1, maybe I'm missing something but I'm not sure why it's there. I also don't understand why the correlations between the climate variables and the PCs are presented in B – this is clearly visible from the eigenvectors in A – but if it is informative I would suggest more discussion.

We removed panel B as suggested by the reviewer. The revision and clarification of associated text was described above in more detail

Figure 3: Very interesting. It might be interesting to see the correlation plot in A separated by men and women.

The updated figure features the following: (A) the correlation plot is now separated by men and women; panel (D) and (E) were amended and now grouped into 10 DAPC clusters.

What happened to the GWAS analysis?

We included a new figure for GWAS as described above

Methods:Plant Materials: seems reasonable and sufficiently described.Sequencing and Variant Calling: I'm not very familiar with RADseq calling pipelines but it seems reasonable. This is a well-established approach though and if the authors are following a previously published processing pipeline it would be good to cite.

We now expanded the RADseq description in the Materials and methods:

“In short, ddRAD is based on a double restriction of the target DNA, typically using a rare cutter and a frequent cutter, followed by sequencing of restriction fragments to reduce the complexity of the target genome.”

Spatial and bioclimatic characterization: It's not clear to me when the different datasets are used, and why you would use one over the other.

We now realize that it was a bad choice to refer to MEMs as spatial characterization and we revised this definition. The section “Spatial and bioclimatic characterization” has been renamed “Bioclimatic characterization". For “spatial” characterization we meant characterization at the sampling locations in terms of geographical positions, from which we derived Moran Eigenvector Map’s, which are non-correlated (distance-based) geographical features of the collection. This set of eigenvectors was then used to build the GF function along with non-collinear bioclimatic variables. The manuscript has been amended accordingly throughout.

470 Why use the highest resolution for something like a landrace that has a km rather than meter resolution?

The climatic resolution used is 2.5 minutes, meaning that we used cells that roughly measure 16 square kilometers (4x4 km) resulting from downscaling of climate models. This is now clarified in text:

“Current climate data (1970-2000 averages) relative to teff landraces’ sampling sites were retrieved from the WorldClim 2 database of global interpolated climate data (Fick and Hijmans, 2017) at the highest available spatial resolution of 2.5’ (approximately 4 x 4 km). “

To use a lower resolution (bigger cells) would have affected our power to distinguish geographically close locations characterized by different bioclimatic features.

More information on the grow-out locations would be welcome.

More information about the test locations were included in the Materials and methods section.

Reviewer #2:In the context of teff production and consumption in Ethiopia the authors advocate breeding initiatives to strengthen this crop that is important for the livelihoods of the many small holders in Ethiopia by highlighting the limited yields, the limited breeding efforts made in this crop and the anticipated impactful climate change. By using genetic, climatic, geographic and participatory variety selection, gene pools are identified that would best inform parents to use for future breeding initiatives. This goes even further and shows the possibility to identify crucial genetic markers needed for breeding focus.With convincing strength, the authors employ genetic analysis of landraces collected all over the country and link this to the different agro-ecological regions and niches as well as the different ethnic traditions and to the climatic data including a forecast of climatic factors based on an extrapolation of historic climate data. Although overwhelming at first, the large amount of figures well illustrate the argument and well satisfy if one wants to get some deeper understanding (supplementary figures). A great innovative strength is the proposition of a standard toolbox to combine the different data to provide product profiles for teff breeding.The authors rightly state that the suitability of varieties does not only depend on ecological, climatic and product data but also on socio-cultural variables and they take this into account by stating that the regions coincide with the different ethnic groups in the country. These ethnic groups could however be named and their differences with regard to teff related practices and preferences highlighted and related to the two different groups of farmers chosen for the participatory variety selection. The selection strategy of the farmers could also be provided.

We revised the text to streamline the definition of the socioeconomic implications on teff distribution. We cannot go as far as defining the ethnicities of the farmers conducting the PVS, although they come from different regions and speak different languages. We tried to convey the point that we do not expect to have differences in evaluation of teff varieties (as shown by the collinearity of farmer evaluations across locations), but rather reduced seed exchange across regions due to social dynamics (local markets) and limited travel capacity. See answer to reviewer#1 for a detailed description of the rationale of grouping accessions by local administrative areas. The description of the farmer selection strategy was included:

“Participating farmers were selected with the help of agricultural officers. They all had experience on teff production and were recognized as a local agricultural expert in teff cultivation. Being local farmers, they spoke different languages (Amharic and Oromo) and local interpreters were employed. Farmers were engaged in focus group discussions prior to the PVS to discuss most relevant traits in teff cultivation and to attend training on how to perform the PVS.”

A little more discussion could be provided on the representativeness of the two PVS and phenotyping sites and the farmers evaluating those.

More information about the sites and their representativeness was given in the Materials and methods section (see answer above)

In the conclusion the authors cite the TRICOT approach as a way forward to decentralize PVS and get better systematic and representative testing of the genetic resources. The limitation of the only two sites in which the genetic collections were grown and the possible G x E influence on the phenotyping of the varieties could also be highlighted. However, the fact that with the two locations such great results were obtained in linking all the datasets shows the potential of the approach taken.

This point is well taken, and we rephrased the conclusion to acknowledge it (including the limitation on GxE):

“Transdisciplinary methods may support the integration of smallholder farmers in modern breeding and agricultural value chains. Modern data-driven research may then efficiently harness the genetic diversity generated in farmers’ fields. Teff is rapidly emerging from the NUS status and is projected towards modern breeding. Genebank genomics may be systematically used to map its large agrobiodiversity (Woldeyohannes et al., 2020), fully disclosing the opportunity to use next generation breeding technologies (Varshney et al., 2021) and large-scale genomic selection (Poland, 2015). Multi-location trials are needed to capture the range of genotype by environment (G x E) interactions that influence agronomic performance of teff, which we could only partially characterize. Decentralized varietal evaluation approaches can be used to scale up the testing of teff genetic resources (de Sousa et al., 2021; van Etten et al., 2019) to better capture G x E, produce new varieties with higher local adaptation, and resulting in higher farmers’ varietal adoption. An extensive socioeconomic characterization of teff cropping, including extensive farmers interviews (Labeyrie et al., 2016), may further unveil the dynamics of teff seed exchange and associated flow of information and knowledge related to local cropping (Occelli et al., 2021). Public sector breeding beyond NUS may further enhance the combination of data-driven research with participatory approaches to improve customer and product profiling to achieve social impact as cost effective as possible.”

321 landraces from 3850 Provide were selected for the study. It will be important to provide the selection strategy used to get to the 321 varieties. This selection strategy could also be highlighted outside of the methodology. The selection strategy might also be related to the time of collection of the varieties. Are these 321 chosen in a way to simply represent the largest genetic diversity in the collection or is the time of collection of the samples also included: to represent the varieties that are currently still cultivated?

An updated discussion of the relevance of breeding materials / landraces was added in text (see previous responses to reviewers). The selection of the core collection was better detailed in the Materials and methods:

“The E. teff diversity panel (EtDP) used in this study was derived from a larger teff collection of 3,850 accessions held at the Ethiopian Biodiversity Institute (EBI) (Addis Abeba, Ethiopia), which represents the world's largest active teff collection and was amplified and characterized by Woldeyohannes and collaborators (Woldeyohannes et al., 2020). Criteria for selecting the EtDP from the larger EBI collection were the following: (i) visible morphological variation for panicle traits, (ii) geographical and agroecological representativeness, (iii) apparent grain yield potential, (iv) presence of different maturity groups, and (v) presence of associated traditional names of specific landraces (e.g. Bunninye, Murri, Fesho). The EtDP includes 321 farmer varieties, sided by all 45 E. teff improved varieties released since the beginning of teff breeding program and until the selection of the EtDP. Improved varieties were obtained by a selection conducted by Ethiopian agricultural research centers. Seven accessions of teff wild relatives E. pilosa and E. curvula were also included in the collection (Supplementary file 1A).”

With regards to choosing men and women and comparing them separately there is not discussion on why one would expect differences. Only in the conclusion Weltzien et al. is cited in this respect but this needs more attention and reference to some more literature, especially in relation to the Ethiopian context. What is especially needed is an explanation of the different tasks (roles) of men and women in Ethiopia with regards to teff cultivation and possibly processing. This should be provided in general and in relation to the two study sites. This could also provide the background to a short discussion to interpret the similarities and the difference observed (appreciation being more related to biomass yield and panicle weight for women while for men it was plant height and grain filling period, could this be explained by women using plant parts for animal feed or any other activity in which they more feature?). I know that this is not the main focus of the paper but the fact that men and women's preferences were analyzed and discussed separately demands a discussion with regards to the results and the strategy used to select the men and women.

An improved description of farmers’ selection was included in the Materials and methods section:

“A participatory variety selection (PVS) was conducted in the two locations, involving 35 experienced teff farmers: 15 men and 10 women in Adet, five men and five women in Akaki. PVS was conducted close to physiological maturity in each location so to maximize variation between plots. Participating farmers were selected with the help of agricultural officers. They all had experience on teff production and were recognized as a local agricultural expert in teff cultivation. Being local farmers, they spoke different languages (Amharic and Oromo) and local interpreters were employed.”

An explanation of gender difference in trait evaluations was added in the Results section, citing relevant literature:

“Both women and men farmers were selected because they were teff growers, and their responses were analyzed separately to highlight distinct criteria for evaluating traits. Previous studies conducted in Ethiopia showed that agricultural traits relevant to marketability, food and drink preparations, animal feed and construction are all perceived differently by women and men farmers (Assefa et al., 2014; Mancini et al., 2017).”

And:

“Different dynamics of trait prioritization by men and women are expected, and may reflect gender roles in the value chain (Weltzien et al., 2019): while men are mostly endowed with field work, women are concerned with marketability of grains (Mancini et al., 2017). However, men and women alike were expert teff growers, as reflected by the consistency of their evaluations (Figure 3).”

This work is not only relevant to under NUS crops but also for breeding in general and especially for public sector breeding. In this respect the authors could open up this approach to be more generally applied to other crops and could highlight the importance of their approach for customer and product profiling and especially for customer profiling with which especially public breeding is tasked to achieve social impact as cost effective as possible.

Thank you for your comment. We reshaped the conclusion section so to reflect a more general interpretation of our results:

“An extensive socioeconomic characterization of teff cropping, including extensive farmers interviews (Labeyrie et al., 2016), may further unveil the dynamics of teff seed exchange and associated flow of information and knowledge related to local cropping (Occelli et al., 2021). Public sector breeding beyond NUS may further enhance the combination of data-driven research with participatory approaches to improve customer and product profiling to achieve social impact as cost effective as possible.”

FiguresWith regards to figures 1 and 2: it is great to have all these figures close to each other for good cross referencing, however in that case it will be very important to have extremely good resolution. A solution could be to arrange the figures in a way that the now smaller figures in Figure 1 become bigger and the larger figure (Figure 1 D) become a little smaller so that the PCs are better visible, this is minor work to arrange. Also, the legenda with the colored dots indicating the DAPC clusters could be made much bigger for easier determination of the colors. The same counts for figure 2.

Figure 1 and Figure 2 were amended as follows:

Figure 1: panel (A) ADMIXTURE results for the pruned SNPs dataset at values of K = 20 grouped according to DAPC clustering results and IBS-based hierarchical clustering (hclust()), Panel (B) and (C) are now larger and the shape of the dots allows to distinguish between teff landraces and varieties. (D) is smaller, while (E) and (F) were kept the same size. The figures were amended and grouped using 10 DAPC clusters instead of 6. The legend is now bigger, we think that the suggestions made the overall results much better summarized, thank you.

Please verify the Y axe on figure 3E. Should this not be PA instead of PL? It was panicle appreciation that was evaluated with the farmers not panicle length?

Thanks for the comment. Good point, it was not consistent with the result section, so now Panicle Appreciation (PA) is shown in 3A.

Reviewer #3:The authors have conducted an interdisciplinary characterization of a valuable teff diversity panel using well-supported methods, except for the selection of genetic clusters. Although the methodology itself is not novel, the interdisciplinary nature of the study sheds light on a wide range of Ethiopian teff germplasm characteristics, from environmental adaption to gender preference.Strengths:The incorporation of farmer's knowledge and gender preferences in the genotypic and phenotypic analysis of the germplasm was well done and nicely presented. This kind of information is absolutely necessary for the improvement and intensification of NUS, especially in smallholder farmer systems. Few germplasm characterization studies, both on NUS and staple crops, have reported such a broad range of characteristics.Weaknesses:Although many of the analyses rely on the differentiation of genetic clusters, the authors have inappropriately selected the optimal number of clusters (K). The authors conducted a cross-validated error analysis of various ADMIXTURE K values and a similar analysis of BIC statistics of various DAPC K values. Figure 1—figure supplement 4 clearly demonstrates that K=20 for ADMIXTURE and K=10 for DAPC, as these values of K have the lowest cross-validated error and BIC, respectively. However, the authors claim to have chosen K=6 because (a) there is a slight flattening of the ADMIXTURE error curve after K=6 and (b) K=6 has the lowest DAPC BIC statistic, neither of which are supported by the results.

This criticism is well taken and was addressed by reconsidering results of the DAPC as well as of ADMIXTURE. Now results are interpreted according to K=10 for the DAPC and K=20 for ADMIXTURE; all figures, including those of the supplement, were amended accordingly

Although the authors reported high broad-sense heritability (H2) for several agronomically important traits, variance components for the terms (genotype, environment, GxE, gender, error, etc.) used to estimate H2 were not shown. A table including H2, variance components, and the numbers of levels and observations for each term is necessary to assess the validity of these results. This information would also shed light on the relative influence of genetics, environment, gender, etc. on trait variation.

This data was included as a separate supplementary table (Supplementary file 1C)

As stated above the methods used to select K=6 as the optimal number of clusters was statistically inappropriate. Although K=20 had the lowest cross-validated ADMIXTURE error, this value seems too high. I would suggest using K=10, as this value of K had the lowest DAPC BIC, showed a more clear flattening in the ADMIXTURE error curve, and is closer to your desired value of K=6. Did K=6 correspond to some prior knowledge on the diversity panel, for example from breeders' knowledge? If so, perhaps this could be stated or incorporated in some way to support your choice of K=6.

To address this and the other reviewer’s remark, we decided to focus on the best interpretation (10 clusters) in the main text and in all parts depending on this interpretation. Every figure in the main text as well as in the supplement were amended accordingly.

[Editors’ note: what follows is the authors’ response to the second round of review.]

The manuscript has been improved but there are some remaining issues that need to be addressed, as outlined below:Essential revisions:1) Further explicate the relative contributions of genotype, environment and GxE for the traits evaluated, per the more detailed suggestions of reviewers 1 and 3. This may be supported by the Oryza research suggested by reviewer 2.

We expanded on the relative contribution of G, E, and G x E in traits determination. To this end, we revised a supplemental table, we added a new supplemental table, and expanded the text in the Results section. See response to reviewers for details

2) Reassess the discussion of the GWAS results in the text and supporting material, particularly DM and PC2_bio. Per comments by reviewer 1, the QQ plots show that these traits are highly impacted by residual population structure and thus subject to an excess of false positives.

We carefully reconsidered GWAS results. In some instances, we believe that residual population structure is not harming the interpretation of the results. We discuss these instances while adding notes of caution on the interpretation of the results. See response to reviewers for details

3) Provide a bit more discussion of gendered roles in agricultural production, taking into account comments by reviewer 2.

We expanded the discussion including new literature and citing the interpretation of the reviewer as speculation.

Reviewer #1:The authors in their responses have addressed most of my concerns and I think the paper reads much more clearly now. I only have two additional comments:With respect to the GWAS results, these can now be evaluated with the inclusion of the QQ plots. All traits except bio14 are somewhat confounded with population structure (genome-wide deviation from expectation of no association, or deviation from the line in the QQ plot) and DM and PC2_bio are very correlated with structure and the QQ plot suggests that neither of these traits have truly associated loci in this study. The authors discuss results from especially DM a great deal and I think the discussion of these results should be reevaluated.

It is true that some residual structure is present in most GWAS scan and that as a result there is an increased chance of type II errors. This is now clarified at the beginning of the GWAS result paragraph:

“Some of the associated traits, notably days to maturity and PC2 of bioclimatic variables, showed some statistical inflation on QQ plots likely contributed by residual population structure (Figure 5—figure supplement 2).”

We tried to compensate this using a stringent multiple testing-corrected statistical threshold, which considers p-values distributions specific for each trait. In the following discussion, we argue that this structure does not hinder the significance of our results, which focus on the highest associations to compensate for inflation. Indeed, p-value inflation is also determined by the local LD at associated loci, and this is the case of the QTL on chr 6A for DM. As visible in the corresponding Manhattan plot, high LD in the region in casing many SNPs to be associated to the QTN, and this contributes to the inflation in the QQ plot. However, the underlying cause is not genome-wide structure but rather localized LD. This is now clarified in text:

“Days to maturity were also associated with a LD block at 20.8 Mb on chromosome 6A that was predictive for the GF model (Figure 4 —figure supplement 2; Figure 4 —figure supplement 3). The corresponding peak, the higher for days to maturity, groups many SNPs (Figure 5—figure supplement 1) and contributes to the inflation observed in the QQ plot for this trait (Figure 5—figure supplement 2).”

When a similar interpretation is not supported, we included a more critical discussion of the associations, e.g. here:

“On chromosome 1A at 32.3 Mb, three QTNs for days to maturity colocalized with a large significance peak for precipitation of driest month (bio14) and PC2 of bioclimatic variables, representing seasonality (Figure 4 —figure supplement 2; Figure 5 —figure supplement 1; Figure 5 —figure supplement 2). Although both days to maturity and PC2 of bioclimatic variables show some statistical inflation, the co-mapping of this locus across different traits reinforces its significance.”

And here:

“These candidate genes have not been yet validated, and their evaluation must be cautious especially for those signals that may confound to background genetic structure (Figure 5—figure supplement 2).“

The authors also state in the conclusion that "Multi-location trials are needed to capture the range of genotype by environment (G x E) interactions that influence agronomic performance of teff, which we could only partially characterize" but they do have two locations and are able to model GxE (although, more locations would certainly lead to better estimates). For a genomic resource paper, it would be very beneficial to report the impact of GxE directly (and how the landraces and improved varieties may differ, as has been found in other crops).

In relation to this and to considerations by Reviewer#3, we decided to expand the discussion on GxE. To this end, we followed the approach suggested by the Editor and presented in one of the papers mentioned by Reviewer#2 and included a new supplementary material providing a synthetic representation of the effect of location (*i.e.* environment), genetics, and gender on trait values. Supplementary Table 1D reports p-values for two-way and three-way ANOVA exploring the interactions in each of the trait. We refer to this in the main text:

“Depending on the trait, we found different effects of genetic background, location, and gender, a proxy of genotype by environment (G x E) interactions in determining teff performance and appreciation in tested locations. Genetic clusters had a significant effect on all traits and location was important for yield and yield components (Supplementary file 1D). Panicle appreciation, plant height, panicle length, culm diameter and panicle weight were affected by the interaction of location and genetic cluster. Gender and gender by location interactions always had significant effects for PVS traits (Supplementary File 1D).”

We also modified Supplementary Table 1C to include a new column reporting the percent variance explained by each component of the model as per suggestion of Reviewer#3.

Reviewer #2:The authors have done a great job in revising the manuscript, which is much easier to read and figures and tables. The organisation has also improved. As far as my expertise allows the authors have also well approved the technical analysis part of the manuscript.The sampling of the PVS participants has been described satisfactorily and the reason why to include gender has been articulated. I can imagine that the authors did not find any literature or resources to explain the difference observed between women and men. I highlighted that a reason could be that women are tasked with animal feeding and look at the plant's vegetative parts also as a resource of animal food while men would not focus on that so much because it might be out of their gendered roles. The authors could quickly consider this issue and see if there is any probability in this being the case or any other reason that could be mentioned to explain the differences observed between women and men. Any little clue or hypothetical phrase would give the gender dimension just a little more depth and clue for further investigation, rather than just saying that differences are to be expected.

To our knowledge, there is no literature on gender-differentiated variety preferences in teff, however several studies explored this factor in other crops. We added a reference to a comprehensive review paper from Jill Cairn’s group at CIMMYT, focusing on maize. Although specific to that crop, it provides an interesting perspective to interpret our findings. The paper is referred here:

“In Ethiopian wheat, men are mostly endowed with field work, while women are concerned with marketability of grains (Mancini et al., 2017). In maize, studies across Africa revealed gender-differentiated trait preferences matching different roles in cropping: women are mostly concerned with use traits (e.g. shellability, milling characteristics, storability) (Voss et al., 2021). However, the underlying causes of gender differentiation in teff trait preference need to be further characterized, also in relation to the multi-purpose of teff cultivation as food and feed.”

The authors nicely made the link to general breeding and the development of customer and product profiles in (public) breeding that are more and more expected to deliver on social impact in line with the sustainable development goalsThe issue I raised with regards to the representativeness of the PVS trial locations has also been resolved satisfactorily.This is now a really strong paper and needs to be published so I will be able to share it as a good example of transdisciplinary data driven research.Please correct on line 161-162: reference is made to two figures (as the word 'and' is written) but only figure 6 is mentioned.

Fixed.

Furthermore, I would like to point the authors to the following publications based on work on the under researched African rice (Oryza glaberrima) and how farmers long trajectories of selection resulted in selecting 'robust' varieties that are adapted to change and dynamics (social as well as climatic) rather than narrow local localities. Any reference to this body of work could be made, although the manuscript stands strong as it is without referring to this body of work.http://journals.plos.org/plosone/article?id=10.1371/journal.pone.0034801http://journals.plos.org/plosone/article?id=10.1371/journal.pone.0085953http://journals.plos.org/plosone/article?id=10.1371/journal.pone.0007335https://link.springer.com/article/10.1007/s10745-012-9528-x

Thank you for suggesting these very valuable reads; it was our mistake to have missed them and we are glad to refer to this literature. We cited two works among those suggested, Mokuwa et al. 2013 and 2014. Tirst in the introduction:

“The agrobiodiversity that farmers maintain is the result of genetic, environmental and societal interactions and has large potential for varietal innovation when farmer preferences are factored in the selection process (Mokuwa et al., 2014).”

Then as methodological approach to answer to Reviewer#1 and Reviewer#2 request to expand on GxE interaction:

“We used an analysis of variance (ANOVA) to describe trait differences conditional to teff genetic groups, locations, and, in the case of PVS, to gender with a procedure similar to that used in Mokuwa et al. (2013).”

Reviewer #3:The authors have done a nice job revising the manuscript using 10 genetic clusters, following the recommendations of the reviewers.I would suggest incorporating the information in Supplementary file 1C into Tables 1 and 2. For example, columns for variance explained by genotype, location, rep, error, etc. could be added to these tables.

The information suggested by the reviewer is very interesting and we decided to enrich Supplementary file 1C with a column reporting percent variance explained by different components to the model. We had considered adding this information to Tables 1-2, but we realized that they would become too complex due to the many different models employed, especially for what concerns PVS data. Conversely, we believe that Supplementary file 1C in the new format allows a clear and comprehensive representation of this information.

I also recommend discussing the results in Supplementary file 1C (or Tables 1-2, if the authors choose to merge this information) further. The experimental design appears to have allowed for high estimation of heritability, which the authors have already stated in the first and latest versions of the manuscript. However, as a plant breeder, I would like to know more about the relative contributions of genotype, environment, GxE on the different traits measured in this germplasm. For example, it is interesting that (a) for BPR, location explains 75% of the variance, while genotype explains a 18% of the variance and there is no GxE effect, which contrasts to (b) CDF, which has large GxE variance (66%) and relatively smaller proportions of the variance are explained by genotype (7%) and location (19%). There are likely some trends/comparisons among traits that can be discussed, such as which traits tend to have high GxE vs traits that have high location effects vs traits with high genetic variance.

In response to this suggestion, we incorporated a new supplementary material (Supplementary file 1C), described in detail in the response to Reviewer#1. We also added a more thorough reference to variance components in the model:

“A significant portion of the variance for production traits could be explained by differences in location, including biomass (75%) and yield (67%). Conversely, phenology was mostly explained by genotype: this is the case of days to heading (92%) and days to maturity (86%) (Supplementary file 1C).”

[Editors’ note: what follows is the authors’ response to the third round of review.]

The manuscript has been improved but there are some remaining issues that need to be addressed, as outlined below:1) Both reviewers 1 and 3 are still concerned with the interpretation of the GWAS. Please see especially reviewer 1's lengthy explanation to help clarify interpretation. Please take these concerns into account when interpreting GWAS results and reporting significant loci.

In response to this suggestion, we incorporated a new supplementary material (Supplementary file 1C), described in detail in the response to Reviewer#1. We also added a more thorough reference to variance components in the model:

“A significant portion of the variance for production traits could be explained by differences in location, including biomass (75%) and yield (67%). Conversely, phenology was mostly explained by genotype: this is the case of days to heading (92%) and days to maturity (86%) (Supplementary file 1C).”

Reviewer #1:The paper reads much better and I am satisfied with the incorporation of GxE analysis. I still have concerns with the interpretation of the GWAS results and would recommend a final copy edit to fix the grammatical errors introduced in the editing process.Statistically significant p-values are detected as deviations from the expectation of no-association based on statistical linkage between the causal association and the tested SNP (there are likely no causal loci tested in this limited marker set). The foundational assumption in GWAS is that these associations will be in local LD based on limited recombination in the region surrounding the causal polymorphism, allowing the researcher to zoom in on the loci underlying trait variation. This is visualized on an QQ plot as most dots on the diagonal of no-association with a small deviation associated with only the most significant SNPs (bio14 is a good example of this). On a Manhattan plot, these will for a localized peak around causal loci, characterized by the extent of the haplotype block associated with the causal polymorphism and the sampled marker density.However, linkage can also exist between causal polymorphism and loci across the genome, even on other chromosomes, because selection is not random or evenly applied across a species or population. When populations differ due to drift processes, and selection is nested within randomly varying genetic structure, GWAS tests for the trait under selection will also find associations with the variants associated with differentiation between populations. For example, one population lives in a cold, highland environment and one lives in a tropical, lowland context. The highland population has a shorter growing season and consequently flowers earlier to ensure that the grain can mature in time before frost. The populations were already a bit different due to distance and chance, but now that the flowering window is non-overlapping for these populations it ensures that gene flow is basically stopped and that the populations will move further apart. If one were to do a GWAS for days to flowering or maturity (or for temperature or growing season variation) across these two populations the resulting QQ and Manhattan plot would look like the ones in the analysis for PC_bio2 or for DM; an early a persistent deviation from the expectation of no association in the QQ that localizes all over the genome in a Manhattan plot. This is not to say that there are not real associations (the peak on 6A is likely associated with a real causal locus for DM), but they are confounded with underlying structure and "statistically significant" associations are no longer a good way to identify truly associated loci. A better way to think of GWAS with these qualities is that the top SNPs are enriched for linked causal associations but any given association is suspect.With this in mind, I would suggest a bit more discussion of confounded structure. Perhaps a way to frame it is in the context of local adaptation, tying in the GF models and variance analysis of GxE for the various traits.331-336: The accepted associations for DM and PC2_bio are still very lax and based on the QQ plots heavily confounded with underlying population structure. This underlying structure is almost certainly driving association with the gradient forest model and I think that the conclusion that this association "support the importance of phenology in teff adaptation and geographic distribution" is reasonable, but not because of genetics underlying days to maturity per se. If the authors were to include a statement to this effect the overall interpretation is reasonable.341-343: Again, when two traits are both heavily confounded with underlying structure it is not surprising that they would colocalize (via the third variable of structure), and I would be especially sceptical of colocalized regions between these traits.349-355: Agree that this peak is likely real because of the very strong local LD. Not convinced of anything else

Thank you for your constructive criticism, thorough explanation and dedication in this revision. Indeed, you have been completely right in pointing out the role of underlying structure in the interpretation of the results. In this revision, we worked to incorporate your comments to the fullest. Also in consideration of Rev#3 comments, we decided to conduct a new association scan for those phenotypes showing inflation of the p values based on a visual assessment of the QQ plots. These included DM and PC2_bio as correctly pointed out. To reduce confounding structure, we run again the GWAS with additional PC covariates and we found that 25 covariates (instead of 10 as in the case of the other traits) could successfully reduce inflation of p values. This can be seen on the QQ plots and Manhattan plots in the revised Figure 5 S1 and Figure 5 S2 attached to the resubmission. The procedure is now reported in Materials and methods:

“The first 10 genetic PCs were used as covariates to account for population structure. The Kinship was estimated using the method implemented by VanRaden (VanRaden, 2008). Both kinship and PCA were calculated using the subset of LD-pruned markers used for population genetics analysis. GWAS was run on bioclimatic variables, agronomic traits, and PVS traits. After the first round of mapping with 10 PCs, individual QQ plot were visually surveyed for inflation in the p-value distributions, as these could be caused by sub-optimal correction for population structure. When inflation was detected, the corresponding GWAS scan was run again using 25 genetic PCs as covariates. QTN were called when association surpassed a multiple testing correction with False Discovery Rate (FDR) of 5% using R/q-value (Storey et al., 2021).”

As you correctly suspected, PC2_bio could not yield “true” associations and was excluded from the results. DM still showed all the relevant QTN but this time without the inflation on Chr 6A. The new correction applied to the models of the problematic traits resulted in a reduction of the number of QTNs. However, the remaining QTN are more solid and relevant, and add value to the manuscript. For this we need to tank you once again for your careful revision. The new GWAS scan results are reported in Table 1E, in this version featuring a new column reporting the number of PC covariates used for each trait. The text was updated to report and discuss the new results, as well as to acknowledge the critical points raised in the revision. We expanded the discussion of the role of confounding structure in the interpretation of the associations, we reframed the discussion of GWAS results in combination with the GF and opted for a more cautious interpretation of the results. Here below I report the most relevant text added to respond to the comments above. All changes are highlighted in text with track change.

In Results and Discussion when introducing GWAS results:

“The GWAS led to the identification of a total of 91 unique quantitative trait nucleotides (QTNs) (Supplementary file 1E). Of these, 43 QTNs were associated with bioclimatic variables at sampling sites. Given the low-density genotyping, we did not expect causal loci in the SNP set, yet it is possible to leverage local LD to target candidate genes in the vicinity of associations. Both in the case of trait values and bioclimatic variables, associations may be confounded by underlying LD structure driven by drift processes contributing to teff landraces’ distribution. Teff genetic clusters (Figure 1) show different distributions in regards to bioclimatic variables and traits values, including phenology, suggesting that the expression of adaptive traits may be confounded by underlying structure (Figure 2—figure supplement 2, Figure 3—figure supplement 2). The GWAS analysis was run with varying numbers of genetic covariates, so to optimize model fit in consideration of trait-genotype covariance. Visual assessment of QQ plots was used as a guidance to interpret significant associations, and a stringent threshold was employed so to minimize Type II errors (Figure 5—figure supplement 2).”

In Results and Discussion when discussing GF:

“We found that the turnover of allele frequencies across the cropping area was best predicted by geographic variables (Moran’s eigenvector map variables, MEM), confirming the importance of non-adaptive processes in teff differentiation and the resulting relevance of structure in interpreting associations.

In Results and Discussion when discussing loci of interest:

“The GF outcome was then linked to GWAS to identify genomic loci associated to climate, agronomic performance, farmers’ preference, and adaptive potential of teff (Figure 4 —figure supplement 2). We found that phenology QTNs were relevant in supporting GF prediction, although they could not suffice in explaining the geographic distribution of teff diversity (Figure 4 —figure supplement 3). The importance of phenology in teff adaptation was already reported (Woldeyohannes et al., 2020), although it may be confounded by underlying population structure. On chromosome 1A at 32.3 Mb, precipitation of driest month (bio14) and seasonality of precipitations (bio15) identify a QTN (Figure 5 —figure supplement 1; Figure 5 —figure supplement 2), whose local LD block harbors 176 gene models, among which four with predicted proteins with high homology with maize and Arabidopsis proteins (Supplementary file 1F).”

Reviewer #3:The authors have done a nice job including more information about GxE, genetic variance, etc.I am still not convinced of the GWA results for DM and PC2_bio, which show a very large deviation from the expected p-value distribution. The abundance of false positives cannot merely be dismissed with the authors' statement that "some of the associated traits…showed some statistical inflation on QQ plots likely contributed by residual population structure". A large proportion of all SNPs tested were deemed significant (assuming that the significance cut-off was approximately -log10p = 3.5) for DM and PC2_bio. I would suggest that the authors use a secondary threshold based on a visual assessment of the QQ plots. For DM, I would suspect that the ~5 SNPs with -log10p > 4.8 are truly significant (QTL on chromosome 6A). Similarly for PC2_bio, the 2 SNPs at the tail of the distribution are likely significant.

Thank you for rising this issue. We used your valuable suggestion of assessing QQ plots visually and decided to re-run GWAS with more covariates for traits showing inflation. We changed the manuscript accordingly, updating supplementary materials thanks to novel analyses as detailed in the response to Rev#1.

With respect to the gender aspects of this manuscript, I would recommend including references from similar work done by the NextGen Cassava project. Here are two examples:https://doi.org/10.1007/s12231-018-9421-7https://doi.org/10.1002/csc2.20152

Thank you for suggesting these papers which strongly relate to our work, we have included both in the manuscript text here:

“Rich NUS agrobiodiversity is still conserved in situ in smallholder agriculture systems, where the selection and cultivation conducted by local farmers resulted in the stabilization of farmer varieties largely untapped by breeding which could provide useful adaptation traits (Iragaba et al., 2020).”

and here:

“Studies across smallholder farming systems in Africa showed that gender-differentiated roles result in different trait preferences: both in cassava (Teeken et al., 2018) and in maize (Voss et al., 2021) women are mostly concerned with use traits, while men prioritize agronomic trats.”